# Koopman-based generalization bound: New aspect for full-rank weights

**Yuka Hashimoto**[1,2], **Sho Sonoda**[2], **Isao Ishikawa**[3,2], **Atsushi Nitanda**[4], **Taiji Suzuki**[5,2]
[1] NTT, [2] RIKEN AIP, [3] Ehime University, [4] A*STAR CFAR, [5] The University of Tokyo

## Abstract

We propose a new bound for generalization of neural networks using Koopman operators. Whereas most of existing works focus on low-rank weight matrices, we focus on full-rank weight matrices. Our bound is tighter than existing norm-based bounds when the condition numbers of weight matrices are small. Especially, it is completely independent of the width of the network if the weight matrices are orthogonal. Our bound does not contradict to the existing bounds but is a complement to the existing bounds. As supported by several existing empirical results, low-rankness is not the only reason for generalization. Furthermore, our bound can be combined with the existing bounds to obtain a tighter bound. Our result sheds new light on understanding generalization of neural networks with full-rank weight matrices, and it provides a connection between operator-theoretic analysis and generalization of neural networks.

## 1 Introduction

Understanding the generalization property has been one of the biggest topics for analyzing neural networks. A major approach for theoretical investigation of this topic is bounding some complexity of networks (Bartlett & Mendelson, 2002; Mohri et al., 2018). Intuitively, a large number of parameters makes the complexity and generalization error large. This intuition has been studied, for example, based on a classical VC-dimension theory (Harvey et al., 2017; Anthony & Bartlett, 2009). However, for neural networks, small generalization error can be achieved even in over-parameterized regimes (Novak et al., 2018; Neyshabur et al., 2019). To explain this behavior, norm-based bounds have been investigated (Neyshabur et al., 2015; Bartlett et al., 2017; Golowich et al., 2018; Neyshabur et al., 2018; Wei & Ma, 2019; 2020; Li et al., 2021; Ju et al., 2022; Weinan E et al., 2022). These bounds do not depend on the number of parameters explicitly. However, they are typically described by the $(p, q)$ norms of the weight matrices, and if the norms are large, these bounds grow exponentially with respect to the depth of the network. Another approach to tackle over-parameterized networks is a compression-based approach (Arora et al., 2018; Suzuki et al., 2020). These bounds explain the generalization of networks by investigating how much the networks can be compressed. The bounds get smaller as the ranks of the weight matrices become smaller. However, low-rankness is not the only reason for generalization. Goldblum et al. (2020) empirically showed that even with high-rank weight matrices, networks generalize well. This implies that if the ranks of the weight matrices are large, the existing compression-based bounds are not always tight.

In this paper, we derive a completely different type of uniform bounds of complexity using Koopman operators, which sheds light on why networks generalize well even when their weights are high- or full-rank matrices. More precisely, let $L$ be the depth, $d_j$ be the width of the $j$th layer, $g$ be the final nonlinear transformation, and $n$ be the number of samples. For $j = 1, \ldots, L$, let $W_j \in \mathbb{R}^{d_j \times d_{j-1}}$ be an injective weight matrix, $s_j > d_j/2$ describes the smoothness of a function space $H_j$ where the Koopman operator is defined. Our results are summarized as follows, where $\|\cdot\|$ is the operator norm, $E_j$ and $G_j$ are factors determined by the activation functions and the range of $W_j$, respectively:

$$\text{Rademacher complexity} \leq O\left(\frac{\|g\|_{H_L}}{\sqrt{n}} \prod_{j=1}^{L} \frac{G_j E_j \|W_j\|^{s_j-1}}{\det(W_j^* W_j)^{1/4}}\right). \tag{1}$$

Surprisingly, the determinant factor tells us that if the singular values of $W_j$ are large, the bound gets small. It is tight when the condition number of $W_j$, i.e., the ratio of the largest and the

smallest singular values, is small. Especially, when $W_j$ is orthogonal, $G_j = 1$ and the factor $\|W_j\|^{s_j-1}/\det(W_j^*W_j)^{1/4}$ reduces to 1. We can interpret that $W_j$ transforms signals in certain directions, which makes it easy for the network to extract features of data. Networks with orthogonal weight matrices have been proposed (Maduranga et al., 2019; Wang et al., 2020; Li et al., 2021). Our bound also justifies the generalization property of these networks.

In addition to providing the new perspective, we can combine our bound with existing bounds. In other words, our bound can be a complement to existing bounds. Goldblum et al. (2020) pointed out that the rank of the weight matrix tends to be large near the input layer, but be small near the output layer. By adopting our bound for lower layers and existing bounds for higher layers, we obtain a tight bound that takes the role of each layer into account. The determinant factors come from Koopman operators. Koopman operator is a linear operator defined by the composition of functions. It has been investigated for analyzing dynamical systems and time-series data generated from dynamical systems (Koopman, 1931; Budišić et al., 2012; Kawahara, 2016; Ishikawa et al., 2018; Klus et al., 2020; Hashimoto et al., 2020; Giannakis & Das, 2020; Brunton et al., 2022). Its theoretical aspects also have been studied (Das et al., 2021; Ikeda et al., 2022a;b; Ishikawa, 2023). Connections between Koopman operators and neural networks have also been discussed. For example, efficient learning algorithms are proposed by describing the learning dynamics of the parameters of neural networks by Koopman operators (Redman et al., 2022; Dogra & Redman, 2020). Lusch et al. (2017) applied neural networks to identifying eigenfunctions of Koopman operators to extract features of dynamical systems. Konishi & Kawahara (2023) applied Koopman operators to analyze equilibrium models. On the other hand, in this paper, we represent the composition structure of neural networks using Koopman operators, and apply them to analyzing the complexity of neural networks from an operator-theoretic perspective.

Our main contributions are as follows:
- We show a new complexity bound, which involves both the norm and determinant of the weight matrices. By virtue of the determinant term, our bound gets small if the condition numbers of the weight matrices get small. Especially, it justifies the generalization property of existing networks with orthogonal weight matrices. It also provides a new perspective about why the networks with high-rank weights generalize well (Section 2, the paragraph after Proposition 4, and Remark 3).
- We can combine our bound with existing bounds to obtain a bound which describes the role of each layer (Subsection 4.4).
- We provide an operator-theoretic approach to analyzing networks. We use Koopman operators to derive the determinant term in our bound (Subsection 3.4 and Subsection 4.1).

We emphasize that our operator-theoretic approach reveals a new aspect of neural networks.

## 2 RELATED WORKS

**Norm-based bounds** Generalization bounds based on the norm of $W_j$ have been investigated in previous studies. These bounds are described typically by the $(p,q)$ matrix norm $\|W_j\|_{p,q}$ of $W_j$ (see the upper part of Table 1). Thus, although these bounds do not explicitly depend on the width of the layers, they tend to be large as the width of the layers becomes large. For example, the $(p,q)$ matrix norm of the $d$ by $d$ identity matrix is $d^{1/p}$. This situation is totally different from our case, where the factor related to the weight matrix is reduced to 1 if it is orthogonal. We can see our bound is described by the spectral property of the weight matrices and tight when the condition number of $W_j$ is small. Bartlett et al. (2017); Wei & Ma (2020); Ju et al. (2022) showed bounds with reference matrices $A_j$ and $B_j$. These bounds allow us to explain generalization property through the discrepancy between the weight matrices and fixed reference matrices. A main difference of these bounds from ours is that the existing bounds only focus on the discrepancy from *fixed reference matrices* whereas our bound is based on the discrepancy of the spectral property from a certain *class of matrices*, which is much broader than fixed matrices. Li et al. (2021) showed a bound described by $|\prod_{j=1}^{L}\|W_j\| - 1|$, the discrepancy between the product of largest singular values of $W_1, \ldots, W_L$ and 1. It is motivated by the covering number $c_{\mathcal{X}}$ of the input space $\mathcal{X}$ and a diameter $\gamma_{\mathbf{x}}$ of the covering set, which depends on the input samples $\mathbf{x} = (x_1, \ldots, x_n)$. This bound somewhat describes the spectral properties of $W_j$. However, they only focus on the largest singular values of $W_j$ and do not take the other singular values into account. On the other hand, our bound is described by how much the whole singular values of $W_j$ differ from its largest singular value.

Table 1: Comparison of our bound to existing bounds.

| Authors | Rate | Type |
|---|---|---|
| Neyshabur et al. (2015) | $\dfrac{2^L \prod_{j=1}^{L} \|W_j\|_{2,2}}{\sqrt{n}}$ | Norm-based |
| Neyshabur et al. (2018) | $\dfrac{L \max_j d_j \prod_{j=1}^{L} \|W_j\|}{\sqrt{n}} \left( \sum_{j=1}^{L} \dfrac{\|W_j\|_{2,2}^2}{\|W_j\|^2} \right)^{1/2}$ | |
| Golowich et al. (2018) | $\left( \prod_{j=1}^{L} \|W_j\|_{2,2} \right) \min\left\{ \dfrac{1}{n^{1/4}}, \sqrt{\dfrac{L}{n}} \right\}$ | |
| Bartlett et al. (2017) | $\dfrac{\prod_{j=1}^{L} \|W_j\|}{\sqrt{n}} \left( \sum_{j=1}^{L} \dfrac{\|W_j^T - A_j^T\|_{2,1}^{2/3}}{\|W_j\|^{2/3}} \right)^{3/2}$ | |
| Wei & Ma (2020) | $\dfrac{(\sum_{j=1}^{L} \kappa_j^{2/3} \min\{L^{1/2}\|W_j - A_j\|_{2,2}, \|W_j - B_j\|_{1,1}\}^{2/3})^{3/2}}{\sqrt{n}}$ | |
| Ju et al. (2022) | $\dfrac{\sum_{j=1}^{L} \theta_j \|W_j - A_j\|_{2,2}}{\sqrt{n}}$ | |
| Li et al. (2021) | $\|\mathbf{x}\| \|\prod_{j=1}^{L} \|W_j\| - 1| + \gamma_{\mathbf{x}} + \sqrt{\dfrac{c_{\mathcal{X}}}{n}}$ | |
| Arora et al. (2018) | $\hat{r} + \dfrac{L \max_i \|f(x_i)\|}{\hat{r}\sqrt{n}} \left( \sum_{j=1}^{L} \dfrac{1}{\mu_j^2 \mu_{j\to}^2} \right)^{1/2}$ | Compression |
| Suzuki et al. (2020) | $\dfrac{\hat{r}}{\sqrt{n}} + \sqrt{\dfrac{L}{n}} \left( \sum_{j=1}^{L} \tilde{r}_j(\tilde{d}_{j-1} + \tilde{d}_j) \right)^{1/2}$ | |
| Ours | $\dfrac{\|g\|_{H_L}}{\sqrt{n}} \prod_{j=1}^{L} \dfrac{G_j E_j \|W_j\|^{s_j - 1}}{\det(W_j^* W_j)^{1/4}}$ | Operator-theoretic |

$\kappa_j$ and $\theta_j$ are determined by the Jacobian and Hessian of the network $f$ with respect to the $j$th layer and $W_j$, respectively. $\tilde{r}_j$ and $\tilde{d}_j$ are the rank and dimension of the $j$th weight matrices for the compressed network.

**Compression-based bounds**  Arora et al. (2018) and Suzuki et al. (2020) focused on compression of the network and showed bounds that also get small as the rank of the weight matrices decreases, with the bias $\hat{r}$ induced by compression (see the middle part of Table 1). Arora et al. (2018) introduced layer cushion $\mu_j$ and interlayer cushion $\mu_{j\to}$ for layer $j$, which tends to be small if the rank of $W_j$ is small. They also observed that noise is filtered by the weight matrices whose stable ranks are small. Suzuki et al. (2020) showed the dependency on the rank more directly. The bounds describe why networks with low-rank matrices generalize well. However, networks with high-rank matrices can also generalize well (Goldblum et al., 2020), and the bounds cannot describe this phenomenon. Arora et al. (2018, Figure 1) also empirically show that noise rapidly decreases on higher layers. Since the noise stability is related to the rank of the weight matrices, the result supports the tightness of their bound on higher layers. However, the result also implies that noise does not really decrease on lower layers, and we need an additional theory that describes what happens on lower layers.

## 3 PRELIMINARIES

### 3.1 NOTATION

For a linear operator $W$ on a Hilbert space, its range and kernel are denoted by $\mathcal{R}(W)$ and $\ker(W)$, respectively. Its operator norm is denoted by $\|W\|$. For a function $p \in L^\infty(\mathbb{R}^d)$, its $L^\infty$-norm is denoted by $\|p\|_\infty$. For a function $h$ on $\mathbb{R}^d$ and a subset $\mathcal{S}$ of $\mathbb{R}^d$, the restriction of $h$ on $\mathcal{S}$ is denoted by $h|_\mathcal{S}$. For $f \in L^2(\mathbb{R}^d)$, we denote the Fourier transform of $f$ as $\hat{f}(\omega) = \int_{\mathbb{R}^d} f(x)e^{-ix\cdot\omega}dx$. We denote the adjoint of a matrix $W$ by $W^*$.

### 3.2 KOOPMAN OPERATOR

Let $\mathcal{F}_1$ and $\mathcal{F}_2$ be function spaces on $\mathbb{R}^{d_1}$ and $\mathbb{R}^{d_2}$. For a function $f : \mathbb{R}^{d_1} \to \mathbb{R}^{d_2}$, we define the Koopman operator from $\mathcal{F}_2$ to $\mathcal{F}_1$ by the composition as follows. Let $\mathcal{D}_f = \{g \in \mathcal{F}_2 \mid g \circ f \in \mathcal{F}_1\}$. The *Koopman operator* $K_f$ from $\mathcal{F}_2$ to $\mathcal{F}_1$ with respect to $f$ is defined as $K_f g = g \circ f$ for $g \in \mathcal{D}_f$.

### 3.3 REPRODUCING KERNEL HILBERT SPACE (RKHS)

As function spaces, we consider RKHSs. Let $p$ be a non-negative function such that $p \in L^1(\mathbb{R}^d)$. Let $H_p(\mathbb{R}^d) = \{f \in L^2(\mathbb{R}^d) \mid \hat{f}/\sqrt{p} \in L^2(\mathbb{R}^d)\}$ be the Hilbert space equipped with the inner product $\langle f, g \rangle_{H_p(\mathbb{R}^d)} = \int_{\mathbb{R}^d} \hat{f}(\omega)\hat{g}(\omega)/p(\omega)d\omega$. We can see that $k_p(x,y) := \int_{\mathbb{R}^d} e^{i(x-y)\cdot\omega}p(\omega)d\omega$

is the reproducing kernel of $H_p(\mathbb{R}^d)$, i.e., $\langle k_p(\cdot, x), f \rangle_{H_p(\mathbb{R}^d)} = f(x)$ for $f \in H_p(\mathbb{R}^d)$. Note that since $H_p(\mathbb{R}^d) \subseteq L^2(\mathbb{R}^d)$, the value of functions in $H_p(\mathbb{R}^d)$ vanishes at infinity.

One advantage of focusing on RKHSs is that they have well-defined evaluation operators induced by the reproducing property. This property makes deriving an upper bound of the Rademacher complexity easier (see, for example, Mohri et al. (2018, Theorem 6.12)).

**Example 1** *Set $p(\omega) = 1/(1 + \|\omega\|^2)^s$ for $s > d/2$. Then, $H_p(\mathbb{R}^d)$ is the Sobolev space $W^{s,2}(\mathbb{R}^d)$ of order $s$. Here, $\|\omega\|$ is the Euclidean norm of $\omega \in \mathbb{R}^d$. Note that if $s \in \mathbb{N}$, then $\|f\|^2_{H_p(\mathbb{R}^d)} = \sum_{|\alpha| \le s} c_{\alpha,s,d} \|\partial^\alpha f\|^2_{L^2(\mathbb{R}^d)}$, where $c_{\alpha,s,d} = (2\pi)^d s!/\alpha!/(s - |\alpha|)!$ and $\alpha$ is a multi index. See Appendix K for more details.*

### 3.4 PROBLEM SETTING

In this paper, we consider an $L$-layer deep neural network. Let $d_0$ be the dimension of the input space. For $j = 1, \ldots, L$, we set the width as $d_j$, let $W_j : \mathbb{R}^{d_{j-1}} \to \mathbb{R}^{d_j}$ be a linear operator, let $b_j : \mathbb{R}^{d_j} \to \mathbb{R}^{d_j}$ be a shift operator defined as $x \mapsto x + a_j$ with a bias $a_j \in \mathbb{R}^{d_j}$, and let $\sigma_j : \mathbb{R}^{d_j} \to \mathbb{R}^{d_j}$ be a nonlinear activation function. In addition, let $g : \mathbb{R}^{d_L} \to \mathbb{C}$ be a nonlinear transformation in the final layer. We consider a network $f$ defined as

$$f = g \circ b_L \circ W_L \circ \sigma_{L-1} \circ b_{L-1} \circ W_{L-1} \circ \cdots \circ \sigma_1 \circ b_1 \circ W_1. \tag{2}$$

Typically, we regard that $W_1$ is composed by $b_1$, $\sigma_1$ to construct the first layer. We sequentially construct the second and third layers, and so on. Then we get the whole network $f$. On the other hand, from operator-theoretic perspective, we analyze $f$ in the opposite direction. For $j = 0, \ldots, L$, let $p_j$ be a density function of a finite positive measure on $\mathbb{R}^{d_j}$. The network $f$ is described by the Koopman operators $K_{W_j} : H_{p_j}(\mathbb{R}^{d_j}) \to H_{p_{j-1}}(\mathbb{R}^{d_{j-1}})$, $K_{b_j}, K_{\sigma_j} : H_{p_j}(\mathbb{R}^{d_j}) \to H_{p_j}(\mathbb{R}^{d_j})$ as

$$f = K_{W_1} K_{b_1} K_{\sigma_1} \cdots K_{W_{L-1}} K_{b_{L-1}} K_{\sigma_{L-1}} K_{W_L} K_{b_L} g.$$

The final nonlinear transformation $g$ is first provided. Then, $g$ is composed by $b_L$ and $W_L$, i.e., the corresponding Koopman operator acts on $g$. We sequentially apply the Koopman operators corresponding to the $(L-1)$th layer, $(L-2)$th layer, and so on. Finally, we get the whole network $f$. By representing the network using the product of Koopman operators, we can bound the Rademacher complexity with the product of the norms of the Koopman operators.

To simplify the notation, we denote $H_{p_j}(\mathbb{R}^{d_j})$ by $H_j$. We impose the following assumptions.

**Assumption 1** *The function $g$ is contained in $H_L$, and $K_{\sigma_j}$ is bounded for $j = 1, \ldots, L$.*

**Assumption 2** *There exists $B > 0$ such that for any $x \in \mathbb{R}^d$, $|k_{p_0}(x, x)| \le B^2$.*

We denote by $F$ the set of all functions in the form of (2) with Assumption 1. As a typical example, if we set $p_j(\omega) = 1/(1 + \|\omega\|^2)^{s_j}$ for $s_j > d_j/2$ and $g(x) = e^{-\|x\|^2}$, then Assumption 1 holds if $K_{\sigma_j}$ is bounded, and $k_{p_0}$ satisfies Assumption 2.

**Remark 1** *Let $g$ be a smooth function which does not decay at infinity, (e.g., sigmoid). Although $H_p(\mathbb{R}^d)$ does not contain $g$, we can construct a function $\tilde{g} \in H_p(\mathbb{R}^d)$ such that $\tilde{g}(x) = g(x)$ for $x$ in a sufficiently large compact region and replace $g$ by $\tilde{g}$ in practical cases. See Remark 6 for details.*

For the activation function, we have the following proposition (c.f. Sawano (2018, Theorem 4.46)).

**Proposition 1** *Let $p(\omega) = 1/(1 + \|\omega\|^2)^s$ for $\omega \in \mathbb{R}^d$ $s \in \mathbb{N}$, and $s > d/2$. If the activation function $\sigma$ has the following properties, then $K_\sigma : H_p(\mathbb{R}^d) \to H_p(\mathbb{R}^d)$ is bounded.*

- *$\sigma$ is $s$-times differentiable and its derivative $\partial^\alpha \sigma$ is bounded for any multi-index $\alpha \in \{(\alpha_1, \ldots, \alpha_d) \mid \sum_{j=1}^d \alpha_j \le s\}$.*
- *$\sigma$ is bi-Lipschitz, i.e., $\sigma$ is bijective and both $\sigma$ and $\sigma^{-1}$ are Lipschitz continuous.*

**Example 2** *We can choose $\sigma$ as a smooth version of Leaky ReLU (Biswas et al., 2022). We will see how we can deal with other activation functions, such as the sigmoid and the hyperbolic tangent in Remark 8.*

**Remark 2** *We have $\|K_\sigma\| \leq \|\det(J\sigma^{-1})\|_\infty \max\{1, \|\partial_1\sigma\|_\infty, \ldots, \|\partial_d\sigma\|_\infty\}$ if $s = 1$ and $\sigma$ is elementwise, where $J\sigma^{-1}$ is the Jacobian of $\sigma^{-1}$. As we will discuss in Appendix B, deriving a tight bound for a larger $s$ is challenging and future work.*

## 4 KOOPMAN-BASED BOUND OF RADEMACHER COMPLEXITY

We derive an upper bound of the Rademacher complexity. We first focus on the case where the weight matrices are invertible or injective. Then, we generalize the results to the non-injective case.

Let $\Omega$ be a probability space equipped with a probability measure $P$. We denote the integral $\int_\Omega s(\omega)\mathrm{d}P(\omega)$ of a measurable function $s$ on $\Omega$ by $\mathrm{E}[s]$. Let $s_1, \ldots, s_n$ be i.i.d. Rademacher variables. For a function class $G$ and $x_1, \ldots, x_n \in \mathbb{R}^d$, we denote the empirical Rademacher complexity by $\hat{R}_n(\mathbf{x}, G)$, where $\mathbf{x} = (x_1, \ldots, x_n)$. We will provide an upper bound of $\hat{R}_n(\mathbf{x}, G)$ using Koopman operators in the following subsections.

### 4.1 BOUND FOR INVERTIBLE WEIGHT MATRICES ($d_j = d$)

In this subsection, we focus on the case $d_j = d$ $(j = 0, \ldots, L)$ for some $d \in \mathbb{N}$ and $W_j$ is invertible for $j = 1, \ldots, L$. This is the most fundamental case. For $C, D > 0$, set a class of weight matrices $\mathcal{W}(C, D) = \{W \in \mathbb{R}^{d \times d} \mid \|W\| \leq C, |\det(W)| \geq D\}$ and a function class $F_{\mathrm{inv}}(C, D) = \{f \in F \mid W_j \in \mathcal{W}(C, D)\}$. We have the following theorem for a bound of Rademacher complexity.

**Theorem 2 (First Main Theorem)** *The Rademacher complexity $\hat{R}_n(\mathbf{x}, F_{\mathrm{inv}}(C, D))$ is bounded as*

$$\hat{R}_n(\mathbf{x}, F_{\mathrm{inv}}(C, D)) \leq \frac{B\|g\|_{H_L}}{\sqrt{n}} \sup_{W_j \in \mathcal{W}(C,D)} \left( \prod_{j=1}^{L} \frac{\|p_j/(p_{j-1} \circ W_j^*)\|_\infty^{1/2}}{|\det(W_j)|^{1/2}} \right) \left( \prod_{j=1}^{L-1} \|K_{\sigma_j}\| \right). \quad (3)$$

By representing the network using the product of Koopman operators, we can get the whole bound by bounding the norm of each Koopman operator. A main difference of our bound from existing bounds, such as the ones by Neyshabur et al. (2015); Golowich et al. (2018) is that our bound has the determinant factors in the denominator in Eq. (3). They come from a change of variable when we bound the norm of the Koopman operators, described by the following inequality:

$$\|K_{W_j}\| \leq \left( \|p_j/p_{j-1} \circ W_j^*\|_\infty / |\det(W_j)| \right)^{1/2}, \quad \|K_{b_j}\| = 1$$

for $j = 1, \ldots, L$. Since the network $f$ in Eq. (2) satisfies $f \in H_0$, using the reproducing property of $H_0$, we derive an upper bound of $\hat{R}_n(\mathbf{x}, F(C, D))$ in a similar manner to that for kernel methods (see, Mohri et al. (2018, Theorem 6.12)).

Regarding the factor $\|p_j/(p_{j-1} \circ W_j^*)\|_\infty$ in Eq. (3), we obtain the following lemma and proposition, which show it is bounded by $\|W_j\|$ and induces the factor $\|W_j\|^{s_j-1}/\det(W_j^*W_j)^{1/4}$ in Eq. (1).

**Lemma 3** *Let $p(\omega) = 1/(1 + \|\omega\|^2)^s$ for $s > d/2$ and $p_j = p$ for $j = 0, \ldots, L$. Then, we have $\|p/(p \circ W_j^*)\|_\infty \leq \max\{1, \|W_j\|^{2s}\}$.*

As a result, the following proposition is obtained by applying Lemma 3 to Theorem 2.

**Proposition 4** *Let $p(\omega) = 1/(1 + \|\omega\|^2)^s$ for $s > d/2$ and $p_j = p$ for $j = 0, \ldots, L$. We have*

$$\hat{R}_n(\mathbf{x}, F_{\mathrm{inv}}(C, D)) \leq \frac{B\|g\|_{H_L}}{\sqrt{n}} \left( \frac{\max\{1, C^s\}}{\sqrt{D}} \right)^L \left( \prod_{j=1}^{L-1} \|K_{\sigma_j}\| \right).$$

**Comparison to existing bounds** We investigate the bound (1) in terms of the singular values of $W_j$. Let $\eta_{1,j} \geq \ldots \geq \eta_{d,j}$ be the singular values of $W_j$ and let $\alpha = s - d/2$. Then, the term depending on the weight matrices in bound (1) is described as $\eta_{1,j}^\alpha \prod_{i=1}^{d} r_{i,j}^{1/2}$, where $r_{i,j} = \eta_{1,j}/\eta_{i,j}$. On the other hand, the existing bound by Golowich et al. (2018) is described as $\eta_{1,j}(\sum_{i=1}^{d} r_{i,j}^{-2})^{1/2}$. Since they are just upper bounds, our bound does not contradict the existing bound. Our bound is tight when the condition number $r_{d,j}$ of $W_j$ is small. The existing bound is tight when $r_{d,j}$ is large. Note that the factors in our bound (1) are bounded below by 1.

**Remark 3** *If a weight matrix $W_j$ is orthogonal, then the factor $\|W_j\|^{s_{j-1}}/\det(W_j)^{1/2}$ in the bound (1) is reduced to 1. On the other hand, existing norm-based bounds are described by the $(p.q)$ matrix norm. The $(p,q)$ matrix norm of the $d$ by $d$ identity matrix is $d^{1/p}$, which is larger than 1. Indeed, limiting the weight matrices to orthogonal matrices has been proposed (Maduranga et al., 2019; Wang et al., 2020; Li et al., 2021). The tightness of our bound in the case of orthogonal matrices implies the advantage of these methods from the perspective of generalization.*

**Remark 4** *There is a tradeoff between $s$ and $B$. We focus on the case where $p_j(\omega) = 1/(1+\|\omega\|^2)^s$. According to Lemma 3, the factor $\|p_j/(p_{j-1} \circ W_j^*)\|_\infty$ becomes small as $s$ approaches to $d/2$. However, the factor $B$ goes to infinity as $s$ goes to $d/2$. Indeed, we have $k_p(x,x) = \int_{\mathbb{R}^d} p(\omega)\mathrm{d}\omega = C\int_0^\infty \frac{r^{d-1}}{(1+r^2)^s}\mathrm{d}r \geq C\int_1^\infty \frac{(r-1)^{d/2-1}}{2r^s}\mathrm{d}r > \frac{C}{4}\int_2^\infty r^{d/2-1-s}\mathrm{d}r = \frac{C}{4}\frac{2^{d/2-s}}{s-d/2}$ for some constant $C > 0$. This behavior corresponds to the fact that if $s = d/2$, the Sobolev space is not an RKHS, and the evaluation operator becomes unbounded.*

## 4.2 BOUND FOR INJECTIVE WEIGHT MATRICES ($d_j \geq d_{j-1}$)

We generalize Theorem 2 to that for injective weight matrices. If $d_j > d_{j-1}$, then $W_j$ is not surjective. However, it can be injective, and we first focus on the injective case. Let $\mathcal{W}_j(C, D) = \{W \in \mathbb{R}^{d_{j-1} \times d_j} \mid d_j \geq d_{j-1}, \|W\| \leq C, \sqrt{\det(W^*W)} \geq D\}$ and $F_{\mathrm{inj}}(C, D) = \{f \in F \mid W_j \in \mathcal{W}_j(C, D)\}$. Let $f_j = g \circ b_L \circ W_L \circ \sigma_{L-1} \circ b_{L-1} \circ W_{L-1} \circ \cdots \circ \sigma_j \circ b_j$ and $G_j = \|f_j|_{\mathcal{R}(W_j)}\|_{H_{p_{j-1}}(\mathcal{R}(W_j))}/\|f_j\|_{H_j}$. We have the following theorem for injective weight matrices.

**Theorem 5 (Second Main Theorem)** *The Rademacher complexity $\hat{R}_n(\mathbf{x}, F_{\mathrm{inj}}(C, D))$ is bounded as*

$$\hat{R}_n(\mathbf{x}, F_{\mathrm{inj}}(C, D))) \leq \frac{B\|g\|_{H_L}}{\sqrt{n}} \sup_{W_j \in \mathcal{W}_j(C,D)} \left( \prod_{j=1}^{L} \frac{\|p_{j-1}/(p_{j-1} \circ W_j^*)\|_{\mathcal{R}(W_j),\infty}^{1/2} G_j}{\det(W_j^*W_j)^{1/4}} \right) \left( \prod_{j=1}^{L-1} \|K_{\sigma_j}\| \right).$$

(4)

Since $W_j$ is not always square, we do not have $\det(W_j)$ in Theorem 2. However, we can replace $\det(W_j)$ with $\det(W_j^*W_j)^{1/2}$. As Lemma A, we have the following lemma about the factor $\|p_j/(p_{j-1} \circ W_j^*)\|_{\mathcal{R}(W_j),\infty}$ in Eq. (4).

**Lemma 6** *Let $p_j(\omega) = 1/(1 + \|\omega\|^2)^{s_j}$ for $s_j > d_j/2$ and for $j = 0, \ldots, L$. Then, we have $\|p_{j-1}/(p_{j-1} \circ W_j^*)\|_{\mathcal{R}(W_j),\infty} \leq \max\{1, \|W_j\|^{2s_{j-1}}\}$.*

Applying Lemma 6 to Theorem 5, we finally obtain the bound (1). Regarding the factor $G_j$, the following lemma shows that $G_j$ is determined by the isotropy of $f_j$.

**Lemma 7** *Let $p_j(\omega) = 1/(1 + \|\omega\|^2)^{s_j}$ for $s_j > d_j/2$ and for $j = 0, \ldots, L$. Let $s_j \geq s_{j-1}$ and $\tilde{H}_j$ be the Sobolev space on $\mathcal{R}(W_j)^\perp$ with $p(\omega) = 1/(1 + \|\omega\|^2)^{s_j-s_{j-1}}$. Then, $G_j \leq G_{j,0}\|f_j|_{\mathcal{R}(W_j)} \cdot f_j|_{\mathcal{R}(W_j)^\perp}\|_{H_j}/\|f_j\|_{H_j}$, where $G_{j,0} = \|f_j\|_{\tilde{H}_j}^{-1}$.*

**Remark 5** *The factor $G_j$ is expected to be small. Indeed, if $f_j(x) = e^{-c\|x\|^2}$ (It is true if $g$ is Gaussian and $j = L$.), $G_j$ gets small as $s_j$ and $d_j$ gets large. Moreover, $G_j$ can alleviate the dependency of $\|K_{\sigma_j}\|$ on $d_j$ and $s_j$. See Appendix C for more details.*

## 4.3 BOUNDS FOR NON-INJECTIVE WEIGHT MATRICES

The bounds in Theorems 2 and 5 are only valid for injective weight matrices, and the bound goes to infinity if they become singular. This is because if $W_j : \mathbb{R}^{d_{j-1}} \to \mathbb{R}^{d_j}$ has singularity, $h \circ W$ for $h \in H_j$ becomes constant along the direction of $\ker(W_j)$. As a result, $h \circ W_j$ is not contained in $H_{j-1}$ and $K_{W_j}$ becomes unbounded. To deal with this situation, we propose two approaches that are valid for non-injective weight matrices, graph-based and weighted Koopman-based ones.

### 4.3.1 GRAPH-BASED BOUND

The first approach to deal with this situation is constructing an injective operator related to the graph of $W_j$. For $j = 1, \ldots, L$, let $r_j = \dim(\ker(W_j))$, $\delta_j = \sum_{k=0}^{j} r_k$, and $\tilde{W}_j$ be defined as $\tilde{W}_j(x_1, x_2) = (W_j x_1, P_j x_1, x_2)$ for $x_1 \in \mathbb{R}^{d_{j-1}}$ and $x_2 \in \mathbb{R}^{\delta_{j-2}}$ for $j \geq 2$ and $\tilde{W}_j x = (W_j x, P_j x)$ for $j = 1$. Here, $P_j$ is the projection onto $\ker(W_j)$. Then, $\tilde{W}_j$ is injective (See Appendix I). Let $\tilde{\sigma}_j : \mathbb{R}^{\delta_j} \to \mathbb{R}^{\delta_j}$ and $\tilde{b}_j : \mathbb{R}^{\delta_j} \to \mathbb{R}^{\delta_j}$ be defined respectively as $\tilde{\sigma}_j(x_1, x_2) = (\sigma_j(x_1), x_2)$ and $\tilde{b}_j(x_1, x_2) = (b_j(x_1), x_2)$ for $x_1 \in \mathbb{R}^{d_j}$ and $x_2 \in \mathbb{R}^{\delta_{j-1}}$. In addition, let $\tilde{g} \in H_{p_L}(\mathbb{R}^{\delta_L})$ be defined as $\tilde{g}(x_1, x_2) = g(x_1)\psi(x_2)$ for $x_1 \in \mathbb{R}^{d_L}$ and $x_2 \in \mathbb{R}^{\delta_{L-1}}$, where $\psi$ is a rapidly decaying function on $\mathbb{R}^{\delta_{L-1}}$ so that $\tilde{g} \in H_{p_L}(\mathbb{R}^{\delta_{L-1}+d_L})$. Consider the following network:

$$
\begin{aligned}
\tilde{f}(x) &= \tilde{g} \circ \tilde{b}_L \circ \tilde{W}_L \circ \cdots \circ \tilde{\sigma}_1 \circ \tilde{b}_1 \circ \tilde{W}_1(x) \\
&= \psi(P_L \sigma_{L-1} \circ b_{L-1} \circ W_{L-1} \circ \cdots \circ \sigma_1 \circ b_1 \circ W_1(x), \ldots, P_2 \sigma_1 \circ b_1 \circ W_1(x), P_1 x) \\
&\quad \cdot g(b_L \circ W_L \circ \cdots \circ \sigma_1 \circ b_1 \circ W_1(x)).
\end{aligned}
\tag{5}
$$

Since $\det(\tilde{W}_j^* \tilde{W}_j) = \det(W_j^* W_j + I)$ and $\|\tilde{W}_j\| = \sqrt{\|I + W_j^* W_j\|}$, we set $\tilde{\mathcal{W}}_j(C, D) = \{W \in \mathbb{R}^{d_{j-1} \times d_j} \mid \sqrt{\|I + W^* W\|} \leq C, \sqrt{\det(W^* W + I)} \geq D\}$ and $\tilde{F}(C, D) = \{\tilde{f} \mid f \in F, W_j \in \tilde{\mathcal{W}}_j(C, D)\}$. Moreover, put $\tilde{H}_L = H_{p_L}(\mathbb{R}^{\delta_L})$. By Theorem 5, we obtain the following bound, where the determinant factor does not go to infinity by virtue of the identity $I$ appearing in $\tilde{W}_j$.

**Proposition 8** *The Rademacher complexity $\hat{R}_n(\mathbf{x}, \tilde{F}(C, D))$ is bounded as*

$$
\hat{R}_n(\mathbf{x}, \tilde{F}(C, D))) \leq \frac{B \|\tilde{g}\|_{\tilde{H}_L}}{\sqrt{n}} \sup_{W_j \in \mathcal{W}_j(C,D)} \left( \prod_{j=1}^{L} \frac{\|p_j / (p_{j-1} \circ \tilde{W}_j^*)\|_{\mathcal{R}(\tilde{W}_j), \infty}^{1/2} \tilde{G}_j}{\det(W_j^* W_j + I)^{1/4}} \right) \left( \prod_{j=1}^{L-1} \|K_{\sigma_j}\| \right).
$$

**Remark 6** *The difference between the networks (5) and (2) is the factor $\psi$. If we set $\psi$ as $\psi(x) = 1$ for $x$ in a sufficiently large region $\Omega$, then $\tilde{f}(x) = f(x)$ for $x \in \Omega$. We can set $\psi$, for example, as a smooth bump function (Tu, 2011, Chapter 13). See Appendix D for the definition. If the data is in a compact region, then the output of each layer is also in a compact region since $\|W_j\| \leq \sqrt{\|I + W_j^* W_j\|} \leq C$. Thus, it is natural to assume that $f$ can be replaced by $\tilde{f}$ in practical cases.*

**Remark 7** *The factor $\|\tilde{g}\|_{\tilde{H}_L}$ grows as $\Omega$ becomes large. This is because $\|\psi\|_{L^2(\mathbb{R}^{\delta_{L-1}})}^2$ becomes large as the volume of $\Omega$ gets large. This fact does not contradict the fact that the bound in Theorem 2 goes to infinity if $W_j$ is singular. More details are explained in Appendix E.*

### 4.3.2 WEIGHTED KOOPMAN-BASED BOUND

Instead of constructing the injective operators in Subsection 4.3.1, we can also use weighted Koopman operators. For $\psi_j : \mathbb{R}^{d_j} \to \mathbb{C}$, define $\tilde{K}_{W_j} h = \psi_j \cdot h \circ W_j$, which is called the weighted Koopman operator. We construct the same network as $\tilde{f}$ in Eq. (5) using $\tilde{K}_{W_j}$.

$$
\begin{aligned}
\tilde{f}(x) &= \tilde{K}_{W_1} K_{b_1} K_{\sigma_1} \cdots \tilde{K}_{W_L} K_{b_L} g(x) \\
&= \psi_1(x)\psi_2(\sigma_1 \circ b_1 \circ W_1(x)) \cdots \psi_L(\sigma_{L-1} \circ b_{L-1} \circ W_{L-1} \circ \cdots \circ \sigma_1 \circ b_1 \circ W_1(x)) \\
&\quad \cdot g(b_L \circ W_L \circ \cdots \circ \sigma_1 \circ b_1 \circ W_1(x)) \\
&= \psi(\sigma_{L-1} \circ b_{L-1} \circ W_{L-1} \circ \cdots \circ \sigma_1 \circ b_1 \circ W_1(x), \ldots, \sigma_1 \circ b_1 \circ W_1(x), x) \\
&\quad \cdot g(b_L \circ W_L \circ \cdots \circ \sigma_1 \circ b_1 \circ W_1(x)),
\end{aligned}
$$

where $\psi(x_1, \ldots, x_L) = \psi_1(x_1) \cdots \psi_L(x_L)$ for $x_j \in \mathbb{R}^{d_{j-1}}$. Let $W_{\mathrm{r},j} = W_j|_{\ker(W_j)^\perp}$ be the restricted operator of $W_j$ to $\ker(W_j)^\perp$. Set $\mathcal{W}_{\mathrm{r},j}(C, D) = \{W \in \mathbb{R}^{d_{j-1} \times d_j} \mid \|W\| \leq C, |\det(W_\mathrm{r})| \geq D\}$ and $F_\mathrm{r}(C, D) = \{f \in F \mid W_j \in \mathcal{W}_{\mathrm{r},j}(C, D)\}$. By letting $\psi_j$ decay in the direction of $\ker(W_j)$, e.g., the smooth bump function, $\tilde{K}_{W_j} h$ for $h \in H_j$ decays in all the directions. We only need to care about $W_{\mathrm{r},j}$, not the whole of $W_j$. Note that $\psi_j$ has the same role as $\psi$ in Eq. (5). If the data is in a compact region, we replace $f$ by $\tilde{f}$, which coincides with $f$ on the compact region. By virtue of the decay property of $\psi_j$, we can bound $\tilde{K}_{W_j}$ even if $W_j$ is singular.

**Proposition 9** *Let $p_j(\omega) = 1/(1 + \|\omega\|^2)^{s_j}$ with $s_j \in \mathbb{N}$, $s_j \geq s_{j-1}$, and $s_j > d_j/2$. Let $\tilde{H}_{j-1} = H_{p_{j-1}}(\ker(W_j))$ and $\psi_j$ be a function satisfying $\psi_j(x) = \psi_{j,1}(x_1)$ for some $\psi_{j,1} \in \tilde{H}_{j-1}$, where $x = x_1 + x_2$ for $x_1 \in \ker(W_j)$ and $x_2 \in \ker(W_j)^\perp$. Moreover, let $G_j = \|f_j\|_{H_{j-1}(\ker(W_j)^\perp)}/\|f_j\|_{H_j}$. Then, we have*

$$\hat{R}_n(\mathbf{x}, F_r(C, D))) \leq \frac{B\|g\|_{H_L}}{\sqrt{n}} \sup_{W_j \in \mathcal{W}_{r,j}(C,D)} \left( \prod_{j=1}^{L} \frac{\|\psi_{j,1}\|_{\tilde{H}_{j-1}} G_j \max\{1, \|W_j\|^{s_{j-1}}\}}{|\det(W_{r,j})|^{1/2}} \right) \left( \prod_{j=1}^{L-1} \|K_{\sigma_j}\| \right).$$

**Remark 8** *Although we focus on the singularity of $K_{W_j}$ and use the weighted Koopman operator with respect to $W_j$, we can deal with the singularity of $K_{\sigma_j}$ in the same manner as $K_{W_j}$, i.e., by constructing the weighted Koopman operator $\tilde{K}_{\sigma_j}$ with respect to $\sigma_j$. For example, the sigmoid and hyperbolic tangent do not satisfy the assumption for $\sigma_j$ stated in Proposition 1. This is because the Jacobian of $\sigma^{-1}$ is not bounded. However, $\tilde{K}_{\sigma_j}$ is bounded by virtue of $\psi_j$.*

**Remark 9** *The norm of $\psi_j$ can be canceled by the factor $G_j$. See Appendix F for more details.*

### 4.4 COMBINING THE KOOPMAN-BASED BOUND WITH OTHER BOUNDS

In this subsection, we show that our Koopman-based bound is flexible enough to be combined with another bound. As stated in Subsection 4.1, the case where our bound is tight differs from the case where existing bounds such as the one by Golowich et al. (2018) are tight. We can combine these bounds to obtain a tighter bound. For $1 \leq l \leq L$, let $F_{1:l}$ be the set of all functions in the form

$$\sigma_l \circ b_l \circ W_l \circ \sigma_{l-1} \circ b_{l-1} \circ W_{l-1} \circ \cdots \circ \sigma_1 \circ b_1 \circ W_1 \tag{6}$$

with Assumption 1, and let $F_{1:l,\text{inj}}(C, D) = \{f \in F_{1:l} \mid W_j \in \mathcal{W}_j(C, D)\}$. For $l \leq L - 1$, consider the set of all functions in $H_l$ which have the form

$$g \circ b_L \circ W_L \circ \sigma_{L-1} \circ b_{L-1} \circ W_{L-1} \circ \cdots \circ \sigma_{l+1} \circ b_{l+1} \circ W_{l+1}$$

and consider any nonempty subset $F_{l+1:L}$ of it. For $l = L$, we set $F_{L+1:L} = \{g\}$. Let $F_{l,\text{comb}}(C, D) = \{f_1 \circ f_2 \mid f_1 \in F_{l+1,L}, \ f_2 \in F_{1:l,\text{inj}}(C, D)\}$. The following proposition shows the connection between the Rademacher complexity of $F_{l,\text{comb}}(C, D)$ and that of $F_{l+1:L}$.

**Proposition 10** *Let $\tilde{\mathbf{x}} = (\tilde{x}_1, \ldots, \tilde{x}_n) \in (\mathbb{R}^{d_l})^n$. Let $v_n(\omega) = \sum_{i=1}^{n} s_i(\omega)k_{p_0}(\cdot, x_i)$, $\tilde{v}_n(\omega) = \sum_{i=1}^{n} s_i(\omega)k_{p_l}(\cdot, \tilde{x}_i)$, $\mathcal{W} = \{(W_1, \ldots, W_l) \mid W_j \in \mathcal{W}_j(C, D)\}$, and $\gamma_n = \|v_n\|_{H_0}/\|\tilde{v}_n\|_{H_l}$. Then,*

$$\hat{R}_n(\mathbf{x}, F_{l,\text{comb}}(C, D)) \leq \sup_{(W_1, \ldots, W_l) \in \mathcal{W}} \prod_{j=1}^{l} \left\| \frac{p_{j-1}}{p_{j-1} \circ W_j^*} \right\|_{\mathcal{R}(W_j),\infty}^{1/2} \frac{G_j\|K_{\sigma_j}\|}{\det(W_j^* W_j)^{1/4}}$$

$$\cdot \left( \hat{R}_n(\tilde{\mathbf{x}}, F_{l+1:L}) + \frac{B}{\sqrt{n}} \inf_{h_1 \in F_{l+1:L}} \mathrm{E}^{\frac{1}{2}} \left[ \sup_{h_2 \in F_{l+1:L}} \left\| h_1 - \frac{\|h_2\|_{H_l}\gamma_n}{\|\tilde{v}_n\|_{H_l}} \tilde{v}_n \right\|_{H_l}^2 \right] \right).$$

The complexity of the whole network is decomposed into the Koopman-based bound for the first $l$ layers and the complexity of the remaining $L - l$ layers, together with a term describing the approximation power of the function class corresponding to $L - l$ layers. Note that Proposition 10 generalizes Theorem 5 up to the multiplication of a constant. Indeed, if $l = L$, we have $\hat{R}_n(\mathbf{x}, F_{l+1,L}) = 0$. In addition, we have $\inf_{h_1 \in F_{l+1:L}} \mathrm{E}^{\frac{1}{2}}[\sup_{h_2 \in F_{l+1:L}} \|h_1 - \|h_2\|\gamma_n \tilde{v}_n/\|\tilde{v}_n\|\|^2] = \mathrm{E}^{\frac{1}{2}}[\|g - \|g\|\gamma_n \tilde{v}_n/\|\tilde{v}_n\|\|^2] \leq \|g\| \mathrm{E}^{\frac{1}{2}}[(1 + \gamma_n)^2]$.

**Remark 10** *We can also combine our Koopman-based approach with the existing "peeling" approach, e.g., by Neyshabur et al. (2015); Golowich et al. (2018) (see Appendix G). Then, for $1 \leq l \leq L$, we obtain a bound such as $O\big((\prod_{j=l+1}^{L} \|W_j\|_{2,2})(\prod_{j=1}^{l} \|W_j\|^{s_j}/\det(W_j^* W_j)^{1/4})\big)$. Typically, in many networks, the width grows sequentially near the input layer, i.e., $d_{j-1} \leq d_j$ for small $j$ and decays near the output layer, i.e., $d_{j-1} \geq d_j$ for large $j$. Therefore, this type of combination is suitable for many practical cases for deriving a tighter bound than existing bounds.*

Proposition 10 and Remark 10 theoretically implies that our Koopman-based bound is suitable for lower layers. Practically, we can interpret that signals are transformed on the lower layers so that its essential information is extracted on the higher layers. This interpretation also supports the result in Figure 1 by Arora et al. (2018). Noise is removed (i.e., signals are extracted) by the higher layers, but it is not completely removed by lower layers. We will investigate the behavior of each layer numerically in Section 5 and Appendix J.2.1.

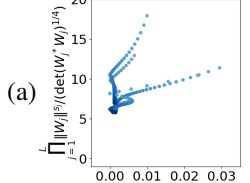 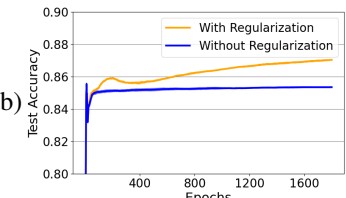 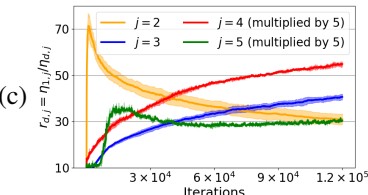

Figure 1: (a) Scatter plot of the generalization error versus our bound (for 5 independent runs). The color is set to get dark as the epoch proceeds. (b) Test accuracy with and without the regularization based on our bound. (c) The condition number $r_{d,j} = \eta_{1,j}/\eta_{d,j}$ of the weight matrix for layer $j = 2, \ldots, 4$.

## 5 NUMERICAL RESULTS

**Validity of the bound** To investigate our bound numerically, we consider a regression problem on $\mathbb{R}^3$, where the target function $t$ is $t(x) = \mathrm{e}^{-\|2x-1\|^2}$. We constructed a simple network $f(x) = g(W_2\sigma(W_1x+b_1)+b_2)$, where $W_1 \in \mathbb{R}^{3\times3}$, $W_2 \in \mathbb{R}^{6\times3}$, $b_1 \in \mathbb{R}^3$, $b_2 \in \mathbb{R}^6$, $g(x) = \mathrm{e}^{-\|x\|^2}$, and $\sigma$ is a smooth version of Leaky ReLU proposed by Biswas et al. (2022). We created a training dataset from samples randomly drawn from the standard normal distribution. Figure 1 (a) illustrates the relationship between the generalization error and our bound $O(\prod_{j=1}^{L} \|W_j\|^{s_j}/(\det(W_j^*W_j)^{1/4}))$. Here, we set $s_j = (d_j+0.1)/2$. In Figure 1 (a), we can see that our bound gets smaller in proportion to the generalization error. In addition, we investigated the generalization property of a network with a regularization based on our bound. We considered the classification task with MNIST. For training the network, we used only $n = 1000$ samples to create a situation where the model is hard to generalize. We constructed a network with four dense layers and trained it with and without a regularization term $\|W_j\| + 1/\det(I + W_j^*W_j)$, which makes both the norm and determinant of $W_j$ small. Figure 1 (b) shows the test accuracy. We can see that the regularization based on our bound leads to better generalization property, which implies the validity of our bound.

**Singular values of the weight matrices** We investigated the difference in the behavior of singular values of the weight matrix for each layer. We considered the classification task with CIFAR-10 and AlexNet (Krizhevsky et al., 2012). AlexNet has five convolutional layers followed by dense layers. For each $j = 2, \ldots, 5$, we computed the condition number $r_{d,j}$ of the weight matrix. The results are illustrated in Figure 1 (c). We scaled the values for $j = 4, 5$ for readability. Since the weight matrix of the first layer ($j = 1$) is huge and the computational cost of computing its singular values is expensive, we focus on $j = 2, \ldots, 5$. We can see that for the second layer ($j = 2$), $r_{d,j}$ tends to be small as the learning process proceeds (as the test accuracy grows). On the other hand, for the third and fourth layers ($j = 3, 4$), the $r_{d,j}$ tends to be large. This means that the behavior of the singular values is different depending on the layers. According to the paragraph after Proposition 4, our bound becomes smaller as $r_{d,j}$ becomes smaller, but the existing bound becomes smaller as $r_{d,j}$ becomes larger. In this case, our bound describes the behavior of the second layer, and the existing bound describes that of the third and fourth layers. See Appendix J for more details and results.

## 6 CONCLUSION AND DISCUSSION

In this paper, we proposed a new uniform complexity bound of neural networks using Koopman operators. Our bound describes why networks with full-rank weight matrices generalize well and justifies the generalization property of networks with orthogonal matrices. In addition, we provided an operator-theoretic approach to analyzing generalization property of neural networks. There are several possible limitations. First, our setting excludes non-smooth activation functions. Generalizing our framework to other function spaces may help us understand these final nonlinear transformations and activation functions. Moreover, although the factor $\|K_{\sigma_j}\|$ is bounded if we set certain activation functions, how this factor changes depending on the choice of the activation function has not been clarified yet. Further investigation of this factor is required. Furthermore, we assume a modification of the structure of the neural network for deriving a bound for non-injective matrices. Thus, we still have room for improvement about this bound. We simply decomposed the product of Koopman operators into every single Koopman operator. For more refined analysis, they should be tied together to investigate the connection between layers. Considering a function space on a manifold that contains the range of the transformation corresponding to each layer may also be effective for resolving this issue. These challenging topics are left to future work.

ACKNOWLEDGEMENT

SS was partially supported by JST PRESTO JPMJPR2125 and JST CREST JPMJCR2015. II was partially supported by JST CREST JPMJCR1913 and JST ACT-X JPMJAX2004. AN was partially supported by JSPS KAKENHI 22H03650 and JST PRESTO JPMJPR1928. TS was partially supported by JSPS KAKENHI 20H00576 and JST CREST.

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

APPENDIX

A  PROOFS

We provide the proofs of the theorems, propositions, and lemmas in the main text.

**Proposition 1**  *Let $p(\omega) = 1/(1 + \|\omega\|^2)^s$ for $\omega \in \mathbb{R}^d$ $s \in \mathbb{N}$, and $s > d/2$. If the activation function $\sigma$ has the following properties, then $K_\sigma : H_p(\mathbb{R}^d) \to H_p(\mathbb{R}^d)$ is bounded.*

- *$\sigma$ is $s$-times differentiable and its derivative $\partial^\alpha \sigma$ is bounded for any multi-index $\alpha \in \{(\alpha_1, \ldots, \alpha_d) \mid \sum_{j=1}^d \alpha_j \le s\}$.*
- *$\sigma$ is bi-Lipschitz, i.e., $\sigma$ is bijective and both $\sigma$ and $\sigma^{-1}$ are Lipschitz continuous.*

**Proof**  For $h \in H(\mathbb{R}^d)$, we have $\|K_\sigma h\|_{H_p(\mathbb{R}^d)}^2 = \sum_{|\alpha| \le s} \|\partial^\alpha (h \circ \sigma)\|_{L^2(\mathbb{R}^d)}^2$. We denote $\sigma(x) = (\sigma_1(x), \ldots, \sigma_d(x))$ and $D_\gamma \sigma(x) = (\partial^\gamma \sigma_1(x), \ldots, \partial^\gamma \sigma_d(x))$ for $\gamma \in \mathbb{N}^d$. By the Faà di Bruno formula, we have

$$\partial^\alpha (h \circ \sigma)(x) = \sum_{|\beta| \le |\alpha|} \partial^\beta h(\sigma(x)) \sum_{i=1}^{|\alpha|} \sum_{\gamma \in p(\alpha,\beta)} \alpha! \prod_{j=1}^i \frac{(\partial^{l_j} \sigma(x))^{k_j}}{k_j! (l_j!)^{|k_j|}},$$

where $p(\alpha, \beta) = \{\gamma = (k_1, \ldots, k_s, l_1, \ldots, l_s) \mid 0 \le l_1 \le \cdots \le l_s, \sum_{j=1}^s k_j = \alpha, \sum_{j=1}^s |k_j| l_j = \beta\}$. Thus, $\partial^\alpha (h \circ \sigma)(x)$ is written as the finite weighted sum of $\partial^\beta h(\sigma(x)) \prod_{i=1}^m (D_{\gamma_i} \sigma(x))^{\delta_i}$ for some $m \le |\alpha|$ and $\beta, \gamma_i, \delta_i \in \mathbb{N}^d$, $|\beta| \le |\alpha|$, $|\gamma_i| \le |\alpha|$, $|\delta_i| \le |\alpha|$. By the boundedness of the derivatives of $\sigma$, there exists $C_{\beta,\gamma,\delta} > 0$ such that

$$\int_{\mathbb{R}^d} \left| \partial^\beta h(\sigma(x)) \prod_{i=1}^m (D_{\gamma_i} \sigma(x))^{\delta_i} \right|^2 \mathrm{d}x \le C_{\beta,\gamma,\delta} \int_{\mathbb{R}^d} |\partial^\beta h(\sigma(x))|^2 \mathrm{d}x.$$

Moreover, by the Lipschitzness of $\sigma^{-1}$, there exists $\tilde{C} > 0$ such that

$$\int_{\mathbb{R}^d} |\partial^\beta h(\sigma(x))|^2 \mathrm{d}x \le \|\det(J\sigma^{-1})\|_\infty \int_{\mathbb{R}^d} |\partial^\beta h(x)|^2 \mathrm{d}x \le \tilde{C} \int_{\mathbb{R}^d} |\partial^\beta h(x)|^2 \mathrm{d}x,$$

where $J\sigma^{-1}$ is the Jacobian of $\sigma^{-1}$, which shows the boundedness of $K_\sigma$. $\qquad\square$

**Theorem 2**  *The Rademacher complexity $\hat{R}_n(\mathbf{x}, F_{\mathrm{inv}}(C, D))$ is bounded as*

$$\hat{R}_n(\mathbf{x}, F_{\mathrm{inv}}(C, D)) \le \frac{B\|g\|_{H_L}}{\sqrt{n}} \sup_{W_j \in \mathcal{W}(C,D)} \left( \prod_{j=1}^L \frac{\|p_j/(p_{j-1} \circ W_j^*)\|_\infty^{1/2}}{|\det(W_j)|^{1/2}} \right) \left( \prod_{j=1}^{L-1} \|K_{\sigma_j}\| \right).$$

We use the following lemma to show Theorem 2.

**Lemma A**  *Assume $W_j : \mathbb{R}^d \to \mathbb{R}^d$ is invertible for $j = 1, \ldots, L$. Then, for $j = 1, \ldots, L$, we have*

$$\|K_{W_j}\| \le \left( \left\| \frac{p_j}{p_{j-1} \circ W_j^*} \right\|_\infty \frac{1}{|\det(W_j)|} \right)^{1/2}, \quad \|K_{b_j}\| = 1.$$

**Proof**  For $h \in H_j$, we have $(\widehat{h \circ W_j})(\omega) = \int_{\mathbb{R}^d} h(W_j x) e^{-\mathrm{i} x \cdot \omega} \mathrm{d}x = \hat{h}(W_j^{-*}\omega)/|\det(W_j)|$. Thus, the norm of the Koopman operator is bounded as

$$\|K_{W_j} h\|_{H_{j-1}}^2 = \int_{\mathbb{R}^d} \frac{|\hat{h}(W_j^{-*}\omega)|^2}{|\det(W_j)|^2 p_{j-1}(\omega)} \mathrm{d}\omega \le \|h\|_{H_j}^2 \sup_{\omega \in \mathbb{R}^d} \left| \frac{p_j(\omega)}{p_{j-1}(W_j^*\omega)} \right| \frac{1}{|\det(W_j)|}.$$

In addition, for $h \in H_j$, we have $\widehat{(h \circ b_j)}(\omega) = \mathrm{e}^{-\mathrm{i}a_j \cdot \omega}\hat{h}(\omega)$. Thus, we obtain $\|K_{b_j}h\|^2 = \|h\|^2$. $\square$

**Proof of Theorem 2** Let $x_1, \ldots, x_n \in \mathbb{R}^{d_0}$ and $s_1, \ldots, s_n$ be i.i.d. Rademacher variables (random variables following the uniform distribution on $\{-1, 1\}$). By the reproducing property of $H_0$ and the Cauchy–Schwartz inequality, we have

$$
\begin{aligned}
\frac{1}{n}\mathrm{E}\bigg[ \sup_{f \in F_{\mathrm{inv}}(C,D)} \sum_{i=1}^{n} s_i f(x_i) \bigg] &= \frac{1}{n}\mathrm{E}\bigg[ \sup_{f \in F_{\mathrm{inv}}(C,D)} \sum_{i=1}^{n} \langle s_i k_{p_0}(\cdot, x_i), f \rangle_{H_0} \bigg] \\
&\leq \frac{1}{n}\mathrm{E}\bigg[ \sup_{f \in F_{\mathrm{inv}}(C,D)} \bigg( \sum_{i,j=1}^{n} s_i s_j k_{p_0}(x_i, x_j) \bigg)^{1/2} \|f\|_{H_0} \bigg] \\
&\leq \frac{1}{n} \sup_{f \in F_{\mathrm{inv}}(C,D)} \|f\|_{H_0} \mathrm{E}^{\frac{1}{2}}\bigg[ \sum_{i,j=1}^{n} s_i s_j k_{p_0}(x_i, x_j) \bigg] \\
&\leq \frac{1}{n} \sup_{f \in F_{\mathrm{inv}}(C,D)} \|f\|_{H_0} \bigg( \sum_{i=1}^{n} k_{p_0}(x_i, x_i) \bigg)^{1/2} \\
&\leq \frac{B}{\sqrt{n}} \sup_{W_j \in \mathcal{W}(C,D)} \|K_{W_1}K_{b_1}K_{\sigma_1} \cdots K_{W_L}K_{b_L}g\|_{H_0} \\
&\leq \frac{B}{\sqrt{n}} \sup_{W_j \in \mathcal{W}(C,D)} \bigg( \prod_{j=1}^{L} \|K_{W_j}\|\|K_{b_j}\|\|K_{\sigma_j}\| \bigg) \|g\|_{H_L},
\end{aligned}
\tag{7}
$$

where the third inequality is derived by the Jensen's inequality. By Lemma A, we obtain the final result. $\square$

**Lemma 3** *Let $p(\omega) = 1/(1 + \|\omega\|^2)^s$ for $s > d/2$ and $p_j = p$ for $j = 0, \ldots, L$. Then, we have $\|p/(p \circ W_j^*)\|_\infty \leq \max\{1, \|W_j\|^{2s}\}$.*

**Proof** By the definition of $p$, we have

$$
\left\| \frac{p}{p \circ W_j^*} \right\|_\infty = \sup_{\omega \in \mathbb{R}^d} \left| \frac{p(\omega)}{p(W_j^*\omega)} \right| = \sup_{\omega \in \mathbb{R}^d} \left| \left( \frac{1 + \|W_j^*\omega\|^2}{1 + \|\omega\|^2} \right)^s \right| \leq \max\{1, \|W_j\|^{2s}\}.
$$

$\square$

**Theorem 5** *Let $\mathcal{W}_j(C,D) = \{W \in \mathbb{R}^{d_{j-1} \times d_j} \mid d_j \geq d_{j-1}, \|W\| \leq C, \sqrt{\det(W^*W)} \geq D\}$ and $F_{\mathrm{inj}}(C,D) = \{f \in F \mid W_j \in \mathcal{W}_j(C,D)\}$. The Rademacher complexity $\hat{R}_n(\mathbf{x}, F_{\mathrm{inj}}(C,D))$ is bounded as*

$$
\hat{R}_n(\mathbf{x}, F_{\mathrm{inj}}(C,D))) \leq \frac{B\|g\|_{H_L}}{\sqrt{n}} \sup_{W_j \in \mathcal{W}_j(C,D)} \bigg( \prod_{j=1}^{L} \frac{\|p_{j-1}/(p_{j-1} \circ W_j^*)\|_{\mathcal{R}(W_j),\infty}^{1/2} G_j}{\det(W_j^*W_j)^{1/4}} \bigg) \bigg( \prod_{j=1}^{L-1} \|K_{\sigma_j}\| \bigg),
$$

*where $G_j = \|f_j|_{\mathcal{R}(W_j)}\|_{H_{P_{j-1}}(\mathcal{R}(W_j))}/\|f_j\|_{H_j}$ and $f_j = g \circ b_L \circ W_L \circ \sigma_{L-1} \circ b_{L-1} \circ W_{L-1} \circ \cdots \circ \sigma_j \circ b_j$.*

**Proof** For $h \in H_j$, we have

$$
(\widehat{h \circ W_j})(\omega) = \int_{\mathbb{R}^{d_{j-1}}} h(W_j x)\mathrm{e}^{-\mathrm{i}x \cdot \omega}\mathrm{d}x = \int_{\mathcal{R}(W_j)} h(x)\mathrm{e}^{-\mathrm{i}x \cdot W_j^{-*}\omega}\mathrm{d}x \frac{1}{|\det(R_j)|} = \frac{\hat{h}(W_j^{-*}\omega)}{|\det(R_j)|},
$$

where $W_j = Q_j R_j$ is the QR decomposition of $W_j$ and $\mathcal{R}(W_j)$ is the range of $W_j$. In addition, we regard $W_j : \mathbb{R}^{d_{j-1}} \to \mathcal{R}(W_j)$. Since $|\det(R_j)| = (\det(R_j^* R_j))^{1/2} = (\det(W_j^* W_j))^{1/2}$, the norm of the Koopman operator is bounded as

$$\|K_{W_j} h\|_{H_{j-1}}^2 = \int_{\mathbb{R}^{d_{j-1}}} \frac{|\hat{h}(W_j^{-*}\omega)|^2}{\det(W_j^* W_j) p_{j-1}(\omega)} \mathrm{d}\omega = \int_{\mathcal{R}(W_j)} \frac{|\hat{h}(\omega)|^2}{\det(W_j^* W_j)^{1/2} p_{j-1}(W_j^*\omega)} \mathrm{d}\omega$$

$$\leq \|h|_{\mathcal{R}(W_j)}\|_{H_{p_{j-1}}(\mathcal{R}(W_j))}^2 \sup_{\omega \in \mathcal{R}(W_j)} \left| \frac{p_{j-1}(\omega)}{p_{j-1}(W_j^*\omega)} \right| \frac{1}{\det(W_j^* W_j)^{1/2}}. \tag{8}$$

Thus, we have

$$\|f\|_{H_0} = \|K_{W_1} f_1\|_{H_0} \leq \|f_1|_{\mathcal{R}(W_1)}\|_{H_{p_0}(\mathcal{R}(W_1))} \sup_{\omega \in \mathcal{R}(W_1)} \left| \frac{p_0(\omega)}{p_0(W_1^*\omega)} \right|^{1/2} \frac{1}{\det(W_1^* W_1)^{1/4}}$$

$$= G_1 \|f_1\|_{H_1} \sup_{\omega \in \mathcal{R}(W_1)} \left| \frac{p_0(\omega)}{p_0(W_1^*\omega)} \right|^{1/2} \frac{1}{\det(W_1^* W_1)^{1/4}}$$

$$\leq G_1 \|K_{\sigma_1}\|_{H_1} \|K_{W_2} f_2\|_{H_1} \sup_{\omega \in \mathcal{R}(W_1)} \left| \frac{p_0(\omega)}{p_0(W_1^*\omega)} \right|^{1/2} \frac{1}{\det(W_1^* W_1)^{1/4}}.$$

Applying the inequality (8) iteratively, we obtain

$$\|f\|_{H_0} \leq \prod_{j=1}^{L} \left\| \frac{p_{j-1}}{p_{j-1} \circ W_j^*} \right\|_{\mathcal{R}(W_j),\infty}^{1/2} \frac{G_j \|K_{\sigma_j}\|}{\det(W_j^* W_j)^{1/4}}. \tag{9}$$

Applying the inequality (9) to $\|f\|_{H_0}$ in the inequality (7) completes the proof. $\qquad\square$

**Lemma 6** *Let $p_j(\omega) = 1/(1 + \|\omega\|^2)^{s_j}$ for $s_j > d_j/2$ and for $j = 0, \ldots, L$. Then, we have $\|p_{j-1}/(p_{j-1} \circ W_j^*)\|_{\mathcal{R}(W_j),\infty} \leq \max\{1, \|W_j\|^{2s_{j-1}}\}$.*

**Proof** By the definition of $p_j$ and the assumption of $s_j \geq s_{j-1}$, we have

$$\left\| \frac{p_{j-1}}{p_{j-1} \circ W_j^*} \right\|_{\mathcal{R}(W_j),\infty} = \sup_{\omega \in \mathcal{R}(W_j)} \left| \frac{p_{j-1}(\omega)}{p_{j-1}(W_j^*\omega)} \right|$$

$$\leq \sup_{\omega \in \mathcal{R}(W_j)} \left| \frac{(1 + \|W_j^*\omega\|^2)^{s_{j-1}}}{(1 + \|\omega\|^2)^{s_{j-1}}} \right|$$

$$\leq \max\{1, \|W_j\|^{2s_{j-1}}\} \sup_{\omega \in \mathcal{R}(W_j)} \left| \left( \frac{1 + \|\omega\|^2}{1 + \|\omega\|^2} \right)^{s_{j-1}} \right|$$

$$= \max\{1, \|W_j\|^{2s_{j-1}}\}.$$

$$\square$$

**Lemma 7** *Let $p_j(\omega) = 1/(1 + \|\omega\|^2)^{s_j}$ for $s_j > d_j/2$ and for $j = 0, \ldots, L$. Let $s_j \geq s_{j-1}$ and $\tilde{H}_j$ be the Sobolev space on $\mathcal{R}(W_j)^\perp$ with $p(\omega) = 1/(1 + \|\omega\|^2)^{s_j - s_{j-1}}$. Then, $G_j \leq G_{j,0} \|f_j|_{\mathcal{R}(W_j)} \cdot f_j|_{\mathcal{R}(W_j)^\perp}\|_{H_j}/\|f_j\|_{H_j}$, where $G_{j,0} = \|f_j\|_{\tilde{H}_j}^{-1}$.*

**Remark 11** *Since $W_j$ is injective, $\dim(\mathcal{R}(W_j)^\perp) = d_j - d_{j-1}$. Thus, we have $s_j - s_{j-1} > \dim(\mathcal{R}(W_j)^\perp)/2$.*

**Proof** By the definition of $G_{j,0}$, and since $\mathcal{R}(W_j)$ and $\mathcal{R}(W_j)^\perp$ are orthogonal, we have

$$\|f_j\|^2_{H_{p_{j-1}}(\mathcal{R}(W_j))} = \int_{\mathcal{R}(W_j)} |\hat{f}_j(\omega)|^2 \frac{1}{p_{j-1}(\omega)} d\omega$$

$$= G^2_{j,0} \int_{\mathcal{R}(W_j)} |\hat{f}_j(\omega_1)|^2 (1 + \|\omega_1\|^2)^{s_{j-1}} d\omega_1 \int_{\mathcal{R}(W_j)^\perp} |\hat{f}_j(\omega_2)|^2 (1 + \|\omega_2\|^2)^{s_j - s_{j-1}} d\omega_2$$

$$= G^2_{j,0} \int_{\mathcal{R}(W_j)} \int_{\mathcal{R}(W_j)^\perp} |\hat{f}_j(\omega_1)|^2 |\hat{f}_j(\omega_2)|^2 \frac{1}{p_j(\omega_1 + \omega_2)} \frac{(1 + \|\omega_1\|^2)^{s_{j-1}}(1 + \|\omega_2\|^2)^{s_j - s_{j-1}}}{(1 + \|\omega_1 + \omega_2\|^2)^{s_j}} d\omega_2 d\omega_1$$

$$= G^2_{j,0} \int_{\mathcal{R}(W_j)} \int_{\mathcal{R}(W_j)^\perp} |\hat{f}_j(\omega_1)|^2 |\hat{f}_j(\omega_2)|^2 \frac{1}{p_j(\omega_1 + \omega_2)}$$
$$\cdot \frac{(1 + \|\omega_1\|^2)^{s_{j-1}}(1 + \|\omega_2\|^2)^{s_j - s_{j-1}}}{(1 + \|\omega_1\|^2 + \|\omega_2\|^2)^{s_{j-1}}(1 + \|\omega_1\|^2 + \|\omega_2\|^2)^{s_j - s_{j-1}}} d\omega_2 d\omega_1$$

$$\le G^2_{j,0} \int_{\mathcal{R}(W_j)} \int_{\mathcal{R}(W_j)^\perp} |\hat{f}_j(\omega_1)\hat{f}_j(\omega_2)|^2 \frac{1}{p_j(\omega_1 + \omega_2)} d\omega_2 d\omega_1$$

$$= G^2_{j,0} \int_{\mathbb{R}^{d_j}} |\widehat{\tilde{f}}_j(\omega)|^2 \frac{1}{p_j(\omega)} d\omega = G^2_{j,0} \|\tilde{f}_j\|^2_{H_j} = G^2_{j,0} \|f_j\|^2_{H_j} \frac{\|\tilde{f}_j\|^2_{H_j}}{\|f_j\|^2_{H_j}},$$

where $\tilde{h}(x) = h(x_1)h(x_2)$ for $x = x_1 + x_2$, $x_1 \in \mathcal{R}(W_j)$, and $x_2 \in \mathcal{R}(W_j)^\perp$. Note that since $\mathcal{R}(W_j)$ and $\mathcal{R}(W_j)^\perp$ are orthogonal, we have $\hat{h}(\omega_1)\hat{h}(\omega_2) = \widehat{\tilde{h}}(\omega)$. $\square$

**Proposition 9** *Let $p_j(\omega) = 1/(1 + \|\omega\|^2)^{s_j}$ with $s_j \in \mathbb{N}$, $s_j \ge s_{j-1}$, and $s_j > d_j/2$. Let $\psi_j$ be a function satisfying $\psi_j(x) = \psi_{j,1}(x_1)$ for some $\psi_{j,1} \in H_{p_{j-1}}(\ker(W_j))$, where $x = x_1 + x_2$ for $x_1 \in \ker(W_j)$ and $x_2 \in \ker(W_j)^\perp$ and $\ker(W_j)$ is the kernel of $W_j$. Let $\mathcal{W}_{r,j}(C, D) = \{W \in \mathbb{R}^{d_{j-1} \times d_j} \mid \|W\| \le C, \; |\det(W_r)| \ge D\}$ and $F_r(C, D) = \{f \in F \mid W_j \in \mathcal{W}_{r,j}(C, D)\}$. Moreover, let $G_j = \|f_j\|_{H_{j-1}(\ker(W_j)^\perp)}/\|f_j\|_{H_j}$. Then, we have*

$$\hat{R}_n(\mathbf{x}, F_r(C, D))) \le \frac{B\|g\|_{H_L}}{\sqrt{n}} \sup_{W_j \in \mathcal{W}_{r,j}(C,D)} \left(\prod_{j=1}^{L} \frac{\|\psi_{j,1}\|_{\tilde{H}_{j-1}} G_j \max\{1, \|W_j\|^{s_{j-1}}\}}{|\det(W_{r,j})|^{1/2}}\right) \left(\prod_{j=1}^{L-1} \|K_{\sigma_j}\|\right).$$

**Proof** For $h \in H_j$, we have $\|\tilde{K}_{W_j} h\|^2_{H_{j-1}} = \sum_{|\alpha| \le s_{j-1}} \|\partial^\alpha(\psi_j \cdot h \circ W_j)\|^2_{L^2(\mathbb{R}^{d_{j-1}})}$, where the directions of the derivatives are along the directions of $\ker(W_j)^\perp$ and $\ker(W_j)$. We have

$$\int_{\mathbb{R}^{d_{j-1}}} |\partial^\beta \psi_j(x) \partial^\gamma(h \circ W_j)(x)|^2 dx \le \|W_j\|^{2|\gamma|} \int_{\mathbb{R}^{d_{j-1}}} |\partial^\beta \psi_j(x)(\partial^\gamma h)(W_j x)|^2 dx. \quad (10)$$

In addition, let $\phi$ be a function satisfying $\phi(x) = \phi_1(x_1)$ for some $\phi_1 \in H_{p_{j-1}}(\ker(W_j))$, where $x = x_1 + x_2$ for $x_1 \in \ker(W_j)$ and $x_2 \in \ker(W_j)^\perp$. Let $u \in H_{p_{j-1}}(\mathbb{R}^{d_{j-1}})$. Then, we have

$$\int_{\mathbb{R}^{d_{j-1}}} |\phi(x)u(W_j x)|^2 dx = \int_{\ker(W_j)} \int_{\ker(W_j)^\perp} |\phi_1(x_1)u(W_j x_2)|^2 dx_2 dx_1$$

$$= \int_{\ker(W_j)} |\phi_1(x_1)|^2 dx_1 \int_{\ker(W_j)^\perp} |u(W_j x_2)|^2 dx_2$$

$$= \|\phi_1\|^2_{L^2(\ker(W_j))} \int_{\ker(W_j)^\perp} |u(W_j x_2)|^2 dx_2. \quad (11)$$

Combining Eqs. (10) and (11), we obtain

$$\int_{\mathbb{R}^{d_{j-1}}} |\partial^\beta \psi_j(x)(\partial^\gamma h \circ W_j)(x)|^2 dx \le \|W_j\|^{2|\gamma|} \|\partial^\beta \psi_j\|^2_{L^2(\ker(W_j))} \int_{\ker(W_j)^\perp} |(\partial^\gamma h)(W_j x)|^2 dx$$

$$\le \|W_j\|^{2|\gamma|} \|\partial^\beta \psi_j\|^2_{L^2(\ker(W_j))} \frac{1}{|\det(W_{r,j})|} \int_{\ker(W_j)^\perp} |\partial^\gamma h(x)|^2 dx.$$

As a result, we have

$$
\begin{aligned}
\|\tilde{K}_{W_j} h\|_{H_{j-1}}^2 &= \sum_{|\alpha| \le s_{j-1}} c_{\alpha, s_{j-1}, d_{j-1}} \|\partial^\alpha (\psi_j \cdot h \circ W_j)\|_{L^2(\mathbb{R}^{d_{j-1}})}^2 \\
&= \sum_{|\alpha| \le s_{j-1}} c_{\alpha, s_{j-1}, d_{j-1}} \int_{\ker(W_j)} \int_{\ker(W_j)^\perp} |\partial^\beta \psi_{j,1}(x_1) \partial^{\alpha-\beta}(h \circ W_j)(x_2)|^2 \mathrm{d}x_2 \mathrm{d}x_1 \\
&= \sum_{|\beta| \le s_{j-1}} \sum_{|\gamma| \le s_{j-1}-|\beta|} c_{\beta, s_{j-1}, r_j} c_{\gamma, s_{j-1}-|\beta|, d_{j-1}-r_j} \int_{\ker(W_j)} \int_{\ker(W_j)^\perp} |\partial^\beta \psi_{j,1}(x_1) \partial^\gamma(h \circ W_j)(x_2)|^2 \mathrm{d}x_2 \mathrm{d}x_1 \\
&\le \sum_{|\beta| \le s_{j-1}} \sum_{|\gamma| \le s_{j-1}-|\beta|} c_{\beta, s_{j-1}, r_j} c_{\gamma, s_{j-1}-|\beta|, d_{j-1}-r_j} \|\partial^\beta \psi_{j,1}\|_{L^2(\ker(W_j))}^2 \frac{\|W_j\|^{2|\gamma|}}{|\det(W_{\mathrm{r},j})|} \|\partial^\gamma h\|_{L^2(\ker(W_j)^\perp)}^2 \\
&\le \frac{\max\{1, \|W_j\|^{2s_{j-1}}\}}{\det(W_{\mathrm{r},j})} \sum_{|\beta| \le s_{j-1}} c_{\beta, s_{j-1}, r_j} \|\partial^\beta \psi_{j,1}\|_{L^2(\ker(W_j))}^2 \sum_{|\gamma| \le s_{j-1}} c_{\gamma, s_{j-1}, d_{j-1}-r_j} \|\partial^\gamma h\|_{L^2(\ker(W_j)^\perp)}^2 \\
&\le \frac{\max\{1, \|W_j\|^{2s_{j-1}}\}}{\det(W_{\mathrm{r},j})} \|\psi_{j,1}\|_{H_{j-1}(\ker(W_j))}^2 \|h\|_{H_{j-1}(\ker(W_j)^\perp)}^2,
\end{aligned}
$$

where $\beta$ in the second line of the above formula is the multi index whose elements corresponding to $\ker(W_j)$ equal to those of $\alpha$ and other elements are zero. In addition, $r_j = \dim(\ker(W_j))$ and $c_{\alpha, s, d} = (2\pi)^d s!/\alpha!/(s - |\alpha|)!$. Therefore, setting $h = f_j$, we have

$$
\|\tilde{K}_{W_j} f_j\|_{H_{j-1}} \le \|\psi_{j,1}\|_{H_{j-1}(\ker(W_j))} \frac{G_j}{|\det(W_{\mathrm{r},j})|^{1/2}} \max\{1, \|W_j\|^{s_{j-1}}\} \|f_j\|_{H_j},
$$

which completes the proof of the proposition. $\square$

**Proposition 10** *Let* $\tilde{\mathbf{x}} = (\tilde{x}_1, \ldots, \tilde{x}_n) \in (\mathbb{R}^{d_l})^n$. *Let* $v_n(\omega) = \sum_{i=1}^n s_i(\omega) k_{p_0}(\cdot, x_i)$, $\tilde{v}_n(\omega) = \sum_{i=1}^n s_i(\omega) k_{p_l}(\cdot, \tilde{x}_i)$, *and* $\gamma_n = \|v_n\|_{H_0} / \|\tilde{v}_n\|_{H_l}$. *Then, we have*

$$
\begin{aligned}
\hat{R}_n(\mathbf{x}, F_{l,\mathrm{comb}}(C, D)) &\le \sup_{\substack{W_j \in \mathcal{W}_j(C,D) \\ (j=1,\ldots,l)}} \prod_{j=1}^l \left\| \frac{p_{j-1}}{p_{j-1} \circ W_j^*} \right\|_{\mathcal{R}(W_j),\infty}^{1/2} \frac{G_j \|K_{\sigma_j}\|}{\det(W_j^* W_j)^{1/4}} \\
&\quad \cdot \left( \hat{R}_n(\tilde{\mathbf{x}}, F_{l+1:L}) + \frac{B}{\sqrt{n}} \inf_{h_1 \in F_{l+1:L}} \mathrm{E}^{\frac{1}{2}} \left[ \sup_{h_2 \in F_{l+1:L}} \left\| h_1 - \frac{\|h_2\|_{H_l} \gamma_n}{\|\tilde{v}_n\|_{H_l}} \tilde{v}_n \right\|_{H_l}^2 \right] \right).
\end{aligned}
$$

**Proof** To simplify the notation, let

$$
\beta_j = \left\| \frac{p_{j-1}}{p_{j-1} \circ W_j^*} \right\|_{\mathcal{R}(W_j),\infty}^{1/2} \frac{G_j \|K_{\sigma_j}\|}{\det(W_j^* W_j)^{1/4}}.
$$

By the reproducing property of $H_0$ and the Cauchy–Shwartz inequality, we have

$$\hat{R}_n(\mathbf{x}, F_{l,\text{comb}}(C,D)) = \frac{1}{n}\mathrm{E}\Bigg[\sup_{f\in F_{l,\text{comb}}(C,D)}\sum_{i=1}^{n}s_i f(x_i)\Bigg] = \frac{1}{n}\mathrm{E}\Bigg[\sup_{f\in F_{l,\text{comb}}(C,D)}\langle v_n, f\rangle_{H_0}\Bigg]$$

$$\leq \frac{1}{n}\mathrm{E}\Bigg[\sup_{f\in F_{l,\text{comb}}(C,D)}\|v_n\|_{H_0}\|K_{W_1}K_{b_1}K_{\sigma_1}\cdots K_{W_l}K_{b_l}K_{\sigma_l}K_{W_{l+1}}K_{b_{l+1}}K_{\sigma_{l+1}}\cdots K_{W_L}K_{b_L}g\|_{H_0}\Bigg]$$

$$\leq \frac{1}{n}\mathrm{E}\Bigg[\sup_{f\in F_{l,\text{comb}}(C,D)}\|v_n\|_{H_0}\prod_{j=1}^{l}\left\|\frac{p_{j-1}}{p_{j-1}\circ W_j^*}\right\|_{\mathcal{R}(W_j),\infty}^{1/2}\frac{G_j\|K_{\sigma_j}\|}{\det(W_j^*W_j)^{1/4}}$$

$$\cdot\|K_{W_{l+1}}K_{b_{l+1}}K_{\sigma_{l+1}}\cdots K_{W_L}K_{b_L}g\|_{H_l}\Bigg]$$

$$\leq \frac{1}{n}\mathrm{E}\Bigg[\sup_{\substack{W_j\in\mathcal{W}_j(C,D)\\(j=1,\ldots,l)}}\prod_{j=1}^{l}\beta_j\sup_{h_2\in F_{l+1:L}}\left\langle\tilde{v}_n,\frac{\|h_2\|_{H_l}\|v_n\|_{H_0}}{\|\tilde{v}_n\|_{H_l}^2}\tilde{v}_n\right\rangle_{H_l}\Bigg]$$

$$= \frac{1}{n}\sup_{\substack{W_j\in\mathcal{W}_j(C,D)\\(j=1,\ldots,l)}}\prod_{j=1}^{l}\beta_j\mathrm{E}\Bigg[\sup_{h_2\in F_{l+1:L}}\left(\langle\tilde{v}_n,h\rangle_{H_l}+\left\langle\tilde{v}_n,\frac{\|h_2\|_{H_l}\gamma_n}{\|\tilde{v}_n\|_{H_l}}\tilde{v}_n-h\right\rangle_{H_l}\right)\Bigg]$$

$$\leq \frac{1}{n}\sup_{\substack{W_j\in\mathcal{W}_j(C,D)\\(j=1,\ldots,l)}}\prod_{j=1}^{L}\beta_j\mathrm{E}\Bigg[\sup_{h_1\in F_{l+1:L}}\langle\tilde{v}_n,h_1\rangle_{H_l}+\sup_{h_2\in F_{l+1:L}}\left\langle\tilde{v}_n,\frac{\|h_2\|_{H_l}\gamma_n}{\|\tilde{v}_n\|_{H_l}}\tilde{v}_n-h\right\rangle_{H_l}\Bigg]$$

$$\leq \sup_{\substack{W_j\in\mathcal{W}_j(C,D)\\(j=1,\ldots,l)}}\prod_{j=1}^{L}\beta_j\Bigg(\hat{R}_n(\mathbf{x},F_{l+1:L})+\frac{1}{n}\mathrm{E}\Bigg[\|\tilde{v}_n\|_{H_l}\sup_{h_2\in F_{l+1:L}}\left\|\frac{\|h_2\|_{H_l}\gamma_n}{\|\tilde{v}_n\|_{H_l}}\tilde{v}_n-h\right\|_{H_l}\Bigg]\Bigg)$$

for any $h\in F_{l+1:L}$. Moreover, again by the Cauchy–Schwartz inequality, we have

$$\mathrm{E}\Bigg[\|\tilde{v}_n\|_{H_l}\sup_{h_2\in F_{l+1:L}}\left\|\frac{\|h_2\|_{H_l}\gamma_n}{\|\tilde{v}_n\|_{H_l}}\tilde{v}_n-h\right\|_{H_l}\Bigg]\leq \mathrm{E}^{\frac{1}{2}}[\|v_n\|_{H_l}^2]\mathrm{E}^{\frac{1}{2}}\Bigg[\sup_{h_2\in F_{l+1:L}}\left\|\frac{\|h_2\|_{H_l}\gamma_n}{\|\tilde{v}_n\|_{H_l}}\tilde{v}_n-h\right\|_{H_l}^2\Bigg]$$

$$\leq B\sqrt{n}\mathrm{E}^{\frac{1}{2}}\Bigg[\sup_{h_2\in F_{l+1:L}}\left\|\frac{\|h_2\|_{H_l}\gamma_n}{\|\tilde{v}_n\|_{H_l}}\tilde{v}_n-h\right\|_{H_l}^2\Bigg],$$

where the second inequality is derived in the same manner as the proof of Theorem 2. Since $h\in F_{l+1:L}$ is arbitrary, we obtain the final result. □

## B  DETAILS OF REMARK 2

To derive a bound $\|K_\sigma\|$, we bound $\sum_{|\alpha|\leq s}c_{\alpha,s,d}\|\partial^\alpha(h\circ\sigma)\|^2$ by $\sum_{|\alpha|\leq s}c_{\alpha,s,d}\|\partial^\alpha h\|^2$. As the proof of the boundedness of $\|K_\sigma\|$, one strategy is using the Faà di Bruno formula.

By the Faà di Bruno formula, we have

$$\partial^\alpha(h\circ\sigma)(x) = \sum_{|\beta|\leq|\alpha|}\partial^\beta h(\sigma(x))\sum_{i=1}^{|\alpha|}\sum_{\gamma\in p(\alpha,\beta)}\alpha!\prod_{j=1}^{i}\frac{(\partial^{l_j}\sigma(x))^{k_j}}{k_j!(l_j!)^{|k_j|}},$$

where $p(\alpha,\beta)=\{\gamma=(k_1,\ldots,k_s,l_1,\ldots,l_s)\mid 0\leq l_1\leq\cdots\leq l_s,\ \sum_{j=1}^{s}k_j=\alpha,\ \sum_{j=1}^{s}|k_j|l_j=\beta\}$. If $\sigma$ is elementwise, $l_j$ and $k_j$ are chosen so that each of them has only one nonzero element, such as $(|l_j|,0,\ldots,0)$. By counting the number of terms in the summation and calculating the coefficients of the terms, we can derive a bound of $\|K_\sigma\|$. However, analytically representing the number of terms in the summation is a challenging task. We admit that this strategy does not give us a tight bound. There may be a more sophisticated approach to deriving a tight bound of $\|K_\sigma\|$. However, the main goal of this paper is to investigate how the property of the weight matrices affects

the generalization property. Since $\|K_\sigma\|$ does not depend on the weight matrices, if we assume the structure of the network is given, the property of the weight matrices does not affect $\|K_\sigma\|$. As we stated in Section 6, investigating $\|K_\sigma\|$ and deriving a tighter bound is future work.

## C    DETAILS OF REMARK 5

We first show that $G_j$ is bounded by a constant that is independent of $f_j$. Since $G_j$ depends on $\ker(W_j)$, we denote it by $G_j(\ker(W_j))$. Let $\mathcal{W}$ be a $k$-dimensional subspace of $\mathbb{R}^d$ and $\{u_1, \ldots, u_k\}$ be an orthonormal basis of $\mathcal{W}$. We consider the average of $G_j(\mathcal{W})$ on the Grassmann manifold $\mathcal{G}_{d,k}$. For this purpose, we fix an orthonormal basis $e_1, \ldots, e_d$ on $\mathbb{R}^d$ and denote by $\partial_i f$ the derivative of $f$ in the direction of $e_i$. In addition, we denote $\partial_U^\alpha = \prod_{j=1}^{k}(\sum_{i=1}^{d} u_{j,i}\partial_i)^{\alpha_j}$, where $u_{j,i} = \langle u_j, e_i \rangle$. Let $s \in \mathbb{N}$. Then, we have

$$
\begin{aligned}
\|f\|_{H^s(\mathcal{W})}^2 &= \sum_{|\alpha|\leq s} c_{\alpha,s,d}\|\partial_U^\alpha f\|_{L^2(\mathcal{W})}^2 \leq \sum_{l=1}^{s}\sum_{|\alpha|=l}\sum_{|\beta|=l} c_{\alpha,s,d}D_{s,d,k}\|\partial^\beta f\|_{L^2(\mathcal{W})}^2 \\
&= D_{s,d,k}\sum_{l=1}^{s}\sum_{|\alpha|=l}c_{\alpha,s,d}\sum_{|\beta|=l}\|\partial^\beta f\|_{L^2(\mathcal{W})}^2 \\
&= D_{s,d,k}\sum_{|\alpha|\leq s}c_{\alpha,s,d}\sum_{|\beta|\leq s}\|\partial^\beta f\|_{L^2(\mathcal{W})}^2 \\
&\leq D_{s,d,k}(2\pi)^d(d+1)^s\sum_{|\beta|\leq s}c_{\beta,s,d}\|\partial^\beta f\|_{L^2(\mathcal{W})}^2.
\end{aligned}
\tag{12}
$$

Here, we used the Cauchy–Schwartz inequality and derive the second inequality as follows:

$$
\begin{aligned}
\left|\prod_{j=1}^{k}\left(\sum_{i=1}^{d}u_{j,i}\partial_i\right)^{\alpha_j}f\right|^2 &\leq \left(\prod_{j=1}^{k}\left(\sum_{i=1}^{d}u_{j,i}^2\right)^{\alpha_j}\right)\left(\prod_{j=1}^{k}\left(\sum_{i=1}^{d}\mathcal{D}_i^2\right)^{\alpha_j}f\right) \\
&= \prod_{j=1}^{k}\left(\sum_{i=1}^{d}\mathcal{D}_i^2\right)^{\alpha_j}f = \prod_{j=1}^{k}\sum_{|\beta|=\alpha_j}\binom{\alpha_j}{\beta}\mathcal{D}_\beta^2 f \leq D_{s,d,k}\sum_{|\beta|=|\alpha|}|\partial^\beta f|^2
\end{aligned}
$$

for some $D_{s,d,k} > 0$ that depends on $s$, $d$, and $k$. Here $\mathcal{D}_i^2$ is the operator defined as $\mathcal{D}_i^2 f = |\partial_i f|^2$ and $\mathcal{D}_\beta^2 f = |\partial_i^{\beta_i} f|^2$. Assume $\{x \in \mathbb{R}^d \mid \|x\| \leq \epsilon\}$ is not contained in the support of $f$. In this case, we have

$$
\sum_{|\beta|\leq s}c_{\beta,s,d}\|\partial^\beta f\|_{L^2(\mathcal{W})}^2 \leq \epsilon^{k-d}\sum_{|\beta|\leq s}c_{\beta,s,d}\int_{\mathcal{W}}|\partial^\beta f(x)|^2|x|^{d-k}\mathrm{d}x
\tag{13}
$$

Integrating the both sides of (13) and by Theorem 2 by Rubin (2018), we have

$$
\int_{\mathcal{G}_{d,k}}\sum_{|\beta|\leq s}c_{\beta,s,d}\|\partial^\beta f\|_{L^2(\mathcal{W})}^2\mathrm{d}\mathcal{W} \leq \epsilon^{k-d}\sum_{|\beta|\leq s}c_{\beta,s,d}\frac{\sigma_d}{\sigma_k}\int_{\mathbb{R}^d}|\partial^\beta f(x)|^2\mathrm{d}x,
$$

where $\sigma_d = 2\pi^{d/2}/\Gamma(d/2)$ and $\Gamma$ is the Gamma function. In addition, $\mathrm{d}\mathcal{W}$ is the integration with respect to the $O(d)$-invariant probability measure on $\mathcal{G}_{d,k}$ and $O(d)$ is the orthogonal group in $\mathbb{R}^d$. Combining with Eq. (12), we obtain

$$
\int_{\mathcal{G}_{d,k}}\|f\|_{H^s(\mathcal{W})}^2\mathrm{d}\mathcal{W} \leq D_{s,d,k}(2\pi)^d(d+1)^s\epsilon^{k-d}\frac{\sigma_d}{\sigma_k}\|f\|_{H^s(\mathbb{R}^d)}^2.
$$

As a result, we obtain

$$
\begin{aligned}
\int_{\mathcal{G}_{d_j,d_{j-1}}}G_j(\mathcal{W})^2\mathrm{d}\mathcal{W} &\leq \int_{\mathcal{G}_{d_j,d_{j-1}}}\frac{\|f\|_{H^{s_j}(\mathcal{W})}^2}{\|f\|_{H_j}^2}\mathrm{d}\mathcal{W} \\
&\leq D_{s_j,d_j,d_{j-1}}(d_j+1)^{s_j}(2\pi)^{d_j}\epsilon^{d_{j-1}-d_j}\frac{\sigma_{d_j}}{\sigma_{d_{j-1}}}.
\end{aligned}
\tag{14}
$$

We admit that this inequality is not tight from the perspective of the dependence on $s_j$, $d_j$, and $d_{j-1}$. However, surprisingly, the inequality (14) shows that the factor $G_j$ is bounded by a constant that is independent of $f_j$ if $\{x \in \mathbb{R}^d \mid \|x\| \leq \epsilon\}$ is not contained in the support of $f_j$. The assumption about $\{x \in \mathbb{R}^d \mid \|x\| \leq \epsilon\}$ can be satisfied if the input is transformed so that it does not take the value near 0.

One of the reasons for the looseness of the above bound is that we upper bounded $\|f\|_{H^{s_{j-1}}(\mathcal{W})}$ by $\|f\|_{H^{s_j}(\mathcal{W})}$. If $f_j$ can be controlled by the Gaussian, then the factor $G_j$ does not seriously affect the bound. Indeed, let $\phi_c(x) = \mathrm{e}^{-\pi^2 \|x\|^2/c}$. In the case of $|\hat{f}_j(\omega_1 + \omega_2)| \geq |\hat{f}_j(\omega_1)|\phi_c(\omega_2)$ for $\omega_1 \in \mathcal{R}(W_j)$ and $\omega_2 \in \mathcal{R}(W_j)^\perp$, we can evaluate $G_j$ as follows:

$$
\begin{aligned}
\|\phi_c\|^2_{H^s(\mathbb{R}^d)} &= \int_{\omega \in \mathbb{R}^d} |\phi_c(\omega)|^2 (1 + \|\omega\|^2)^s \mathrm{d}\omega = \int_{\omega \in \mathbb{R}^d} \mathrm{e}^{-2\pi^2 \|\omega\|^2/c} (1 + \|\omega\|^2)^s \mathrm{d}\omega \\
&= \int_0^\infty \mathrm{e}^{-2\pi^2 r^2/c} (1 + r^2)^s r^{d-1} \mathrm{d}r \cdot 2\pi \prod_{i=1}^{d-2} \tilde{c}_i \\
&= 2\pi \int_0^\infty \mathrm{e}^{-2\pi^2 r^2/c} \sum_{i=0}^s \binom{s}{i} r^{2i+d-1} \mathrm{d}r \prod_{i=1}^{d-2} \tilde{c}_i \\
&= 2\pi \sum_{i=0}^s \binom{s}{i} \int_0^\infty \mathrm{e}^{-t} t^{i+(d-1)/2} \left(\frac{c}{2\pi^2}\right)^{i+(d-1)/2} \frac{\sqrt{c}}{\pi\sqrt{8t}} \mathrm{d}t \prod_{i=1}^{d-2} \tilde{c}_i \\
&\sim c^{s+d/2} \pi^{-2s-d+1} 2^{-s-d/2} \int_0^\infty \mathrm{e}^{-t} t^{s+d/2-1} \mathrm{d}t \prod_{i=1}^{d-2} \tilde{c}_i \\
&= c^{s+d/2} \pi^{-2s-d+1} 2^{-s-d/2} \Gamma(s + d/2) \prod_{i=1}^{d-2} \tilde{c}_i,
\end{aligned}
\tag{15}
$$

where $a \sim b$ means $a/b \to 1$ as $s \to \infty$ and $d \to \infty$. In addition, $\tilde{c}_i = \int_0^\pi \sin^i \theta \mathrm{d}\theta$. Thus, we have

$$
\begin{aligned}
G_j^2 &= \frac{\|f_j|_{\mathcal{R}(W_j)}\|^2_{H_{p_{j-1}}(\mathcal{R}(W_j))}}{\|f_j\|^2_{H_j}} = \frac{\int_{\mathcal{R}(W_j)} |\hat{f}_j(\omega_1)|^2 (1 + \|\omega_1\|^2)^{s_{j-1}} \mathrm{d}\omega_1}{\int_{\mathbb{R}^d} |\hat{f}_j(\omega_1 + \omega_2)|^2 (1 + \|\omega_1\|^2 + \|\omega_2\|^2)^{s_j} \mathrm{d}\omega} \\
&\leq \frac{\int_{\mathcal{R}(W_j)} |\hat{f}_j(\omega_1)|^2 (1 + \|\omega_1\|^2)^{s_{j-1}} \mathrm{d}\omega_1}{\int_{\mathbb{R}^d} |\hat{f}_j(\omega_1)|^2 \phi_c(\omega_2)^2 (1 + \|\omega_1\|^2 + \|\omega_2\|^2)^{s_j} \mathrm{d}\omega} \\
&\leq \frac{\int_{\mathcal{R}(W_j)} |\hat{f}_j(\omega_1)|^2 (1 + \|\omega_1\|^2)^{s_{j-1}} \mathrm{d}\omega_1}{\int_{\mathbb{R}^d} |\hat{f}_j(\omega_1)|^2 \phi_c(\omega_2)^2 (1 + \|\omega_1\|^2)^{s_{j-1}} (1 + \|\omega_2\|^2)^{\tilde{s}_j} \mathrm{d}\omega} \\
&= \frac{1}{\int_{\mathcal{R}(W_j)^\perp} \phi_c(\omega_2)^2 (1 + \|\omega_2\|^2)^{\tilde{s}_j} \mathrm{d}\omega} \\
&\sim \left( c^{\tilde{s}_j + \tilde{d}_j/2} \pi^{-2\tilde{s}_j - \tilde{d}_j + 1} 2^{-\tilde{s}_j - \tilde{d}_j/2} \Gamma(\tilde{s}_j + \tilde{d}_j/2) \prod_{i=1}^{\tilde{d}_j - 2} \tilde{c}_i \right)^{-1},
\end{aligned}
$$

where $\tilde{s}_j = s_j - s_{j-1}$ and $\tilde{d}_j = d_j - d_{j-1}$. Note that since $s_j \geq s_{j-1}$ and $d_j \geq d_{j-1}$, $G_j$ becomes small as $c$ becomes large and $s_j$ and $d_j$ becomes large. The assumption $|\hat{f}_j(\omega_1 + \omega_2)| \geq |\hat{f}_j(\omega_1)|\phi_c(\omega_2)$ means that $\hat{f}_j$ decays slower or equal to the speed of the Gaussian in the direction of $\omega_2$. Even if $c$ is chosen small to satisfy the condition, the factor $\Gamma(\tilde{s}_j + \tilde{d}_j/2)$ becomes sufficiently large if $s_j$ is sufficiently large. As a result, the upper bound becomes sufficiently small if $s_j$ is sufficiently large.

Moreover, the factor $G_j$ can alleviate the dependency of $\|K_{\sigma_j}\|$ on $d_j$ and $s_j$. Even if the dependence of $\|K_{\sigma_j}\|$ on $d_j$ and $s_j$ is exponential, since the exponents appeared in the above evaluation are $-(d_j - d_{j-1})$ and $-(s_j - s_{j-1})$, we can expect that the dependency on $d_j$ and $s_j$ is reduced to the dependency on $d_{j-1}$ and $s_{j-1}$.

## D   DETAILS OF REMARK 6

As an example of $\psi$, we can use a bump function $\psi(x) = 1 - g((\|x\|^2 - a^2)/(b^2 - a^2))$ on $\mathbb{R}^d$ for $0 < a < b$, where $g(x) = f(x)/(f(x) - f(1-x))$, $f(x) = e^{-1/x}$ for $x > 0$, and $f(x) = 0$ for $x \leq 0$. In this case, the support of $\psi$ is $\{x \in \mathbb{R}^d \mid \|x\| \leq b\}$ and $\psi(x) = 1$ for $x \in \{x \in \mathbb{R}^d \mid \|x\| \leq a\}$. If the output of each layer is bounded on $\{x \in \mathbb{R}^d \mid \|x\| \leq a\}$, then we can obtain a modified network that is exactly the same on $\{x \in \mathbb{R}^d \mid \|x\| \leq a\}$ with this bump function. If $a$ and $b$ are large, then the support of $\psi$ becomes large, which makes the $L^2$-norm of $\psi$ large, and the Sobolev norm of $\psi$ also becomes large. If $a - b$ is small, then $|\psi(x) - \psi(y)|/\|x - y\|$ for $\|x\|^2 = a$ and $y = (b/a)x$ becomes large. Thus, the $L^2$-norms of the derivatives of $\psi$ are expected to be large, and the Sobolev norm of $\psi$ also becomes large if $a - b$ is small.

## E   DETAILS OF REMARK 7

The factor $\|\tilde{g}\|_{\tilde{H}_L}$ grows as $\Omega$ becomes large, where $\Omega$ is the region such that $\tilde{f}(x) = f(x)$ on $x \in \Omega$ for the network $f$ and the modified network $\tilde{f}$. Indeed, if $p_L(\omega) = 1/(1 + \|\omega\|^2)^{s_L}$ for $s_L \in \mathbb{N}$, then we have

$$
\begin{aligned}
\|\tilde{g}\|_{\tilde{H}_L}^2 &= \sum_{|\alpha| \leq s_L} c_{\alpha, s_L, \delta_{L-1} + d_L} \|\partial^\alpha(\psi \cdot g)\|_{L^2(\mathbb{R}^{\delta_L})}^2 \\
&= \sum_{|\alpha| \leq s_L} c_{\alpha, s_L, \delta_{L-1} + d_L} \|\partial^\beta \psi \partial^{\alpha - \beta} g\|_{L^2(\mathbb{R}^{\delta_{L-1} + d_L})}^2 \\
&= \sum_{|\alpha| \leq s_L} c_{\alpha, s_L, \delta_{L-1} + d_L} \|\partial^\beta \psi\|_{L^2(\mathbb{R}^{\delta_{L-1}})}^2 \|\partial^{\alpha - \beta} g\|_{L^2(\mathbb{R}^{d_L})}^2,
\end{aligned}
$$

where $\beta$ is the multi index whose elements corresponding to $\delta_{L-1}$ equals to those of $\alpha$ and the other elements are zero. The factor $\|\psi\|_{L^2(\mathbb{R}^{\delta_{L-1}})}^2$ becomes large as the volume of $\Omega$ gets large. Thus, under the condition that $\|\partial^\alpha \psi\|_{L^2(\mathbb{R}^{\delta_{L-1}})}$ does not change, $\|\tilde{g}\|_{\tilde{H}_L}$ becomes large as $\Omega$ gets large. Indeed, assume $\delta_L = 1$, $\psi(x) = 1$ for $x \in \Omega$, and $\Omega$ is an interval (e.g., $\psi$ is the bump function defined in Appendix D). Then we can create a new function $\tilde{\psi}$ that satisfies $\|\tilde{\psi}^{(i)}\|_{L^2(\mathbb{R})} = \|\psi^{(i)}\|_{L^2(\mathbb{R})}$ for any $i = 1, 2, \ldots$, from $\psi$ as follows. Here, $\psi^{(i)}$ is the $i$th derivative of $\psi$. For simplicity, we consider the case where $\Omega = [-a, a]$ for some $a > 0$. For $c > 0$, we set $\tilde{\psi}(x) = 1$ for $x \in [0, c]$, $\tilde{\psi}(x) = \psi(x - c)$ for $x \in (c, \infty)$, $\tilde{\psi}(x) = 1$ for $x \in [-c, 0]$, $\tilde{\psi}(x) = \psi(x + c)$ for $x \in (-\infty, 0)$.

## F   DETAILS OF REMARK 9

In Proposition 9, by combining with the factor $G_j$, the norm of $\psi_j$ can be canceled as follows. Let $p_j(\omega) = 1/(1 + \|\omega\|^2)^{s_j}$ and $s_j = 2s_{j-1}$. We have

$$
\begin{aligned}
&\|\psi_j\|_{H_{j-1}(\ker(W_j))}^2 \|f_j\|_{H_{j-1}(\ker(W_j)^\perp)}^2 \\
&= \int_{\ker(W_j)} |\hat{\psi}_j(\omega_1)|^2 (1 + \|\omega_1\|^2)^{s_{j-1}} d\omega_1 \int_{\ker(W_j)^\perp} |\hat{f}_j(\omega_2)|^2 (1 + \|\omega_2\|^2)^{s_{j-1}} d\omega_2 \\
&\leq \int_{\ker(W_j)} \int_{\ker(W_j)^\perp} |\hat{\psi}_j(\omega_1) \hat{f}_j(\omega_2)|^2 (1 + \|\omega_1 + \omega_2\|^2)^{s_j} d\omega_1 d\omega_2 \\
&= \|\psi_{j,1} f_j|_{\ker(W_j)^\perp}\|_{H_j(\mathbb{R}^{d_{j-1}})}^2.
\end{aligned}
$$

Thus, we obtain

$$
\begin{aligned}
\|\psi_j\|_{H_{j-1}(\ker(W_j))} G_j &= \frac{\|\psi_j\|_{H_{j-1}(\ker(W_j))} \|f_j\|_{H_{j-1}(\ker(W_j)^\perp)}}{\|f_j\|_{H_j}} \\
&\leq \frac{\|\psi_{j,1} f_j|_{\ker(W_j)^\perp}\|_{H_j(\mathbb{R}^{d_{j-1}})}}{\|f_j\|_{H_j}}.
\end{aligned}
$$

If $\psi_{j,1} f_j|_{\ker(W_j)^\perp} = f_j$, e.g. $\psi_{j,1}$ and $f_j$ are the Gaussian, and $d_{j-1} = d_j$, then the factor $\|\psi_j\|_{H_{j-1}(\ker(W_j))} G_j$ is bounded by 1.

## G  DETAILS OF REMARK 10

We can combine our Koopman-based approach with the existing "peeling" approach. For $1 \leq l \leq L$, let $F_l^{\mathrm{ReLU}}(C_1)$ be the set of $l$-layer ReLU networks where the Frobenius norms of $W_1, \ldots, W_l$ are bounded by $C_1$, considered by Neyshabur et al. (2015). Let $\tilde{F}_{1:l}$ be the set of functions defined in the same manner as Eq. (6), except for replacing $\sigma_l$ with $g \in H_l$. In addition, let $\tilde{F}_{1:l,\mathrm{inj}}(C_2, D) = \{f \in \tilde{F}_{1:l} \mid W_j \in \mathcal{W}_j(C_2, D)\}$. We combine $F_{L-l}^{\mathrm{ReLU}}(C_1)$ and $\tilde{F}_{1:l,\mathrm{inj}}(C_2, D)$, and we define $F_{1:l,\mathrm{inj}}^{\mathrm{ReLU},L}(C_1, C_2, D) = \{h \circ f \mid h \in F_{L-l}^{\mathrm{ReLU}}(C_1), \ f \in \tilde{F}_{1:l,\mathrm{inj}}(C_2, D)\}$. Then, applying Theorem 1 by Neyshabur et al. (2015), we can bound the complexity of $L$-layer networks using that of $(L-1)$-layer networks and the Frobenius norm of $W_L$. For example, by Eq. (8) by Golowich et al. (2018)), we have

$$\hat{R}_n(\mathbf{x}, F_{1:l,\mathrm{inj}}^{\mathrm{ReLU},L}(C_1, C_2, D)) \leq \|W_L\|_{2,2} \hat{R}_n\big(\mathbf{x}, \sigma\big(F_{1:l,\mathrm{inj}}^{\mathrm{ReLU},L-1}(C_1, C_2, D)\big)\big)$$
$$\leq 2\|W_L\|_{2,2} \hat{R}_n(\mathbf{x}, F_{1:l,\mathrm{inj}}^{\mathrm{ReLU},L-1}(C_1, C_2, D)),$$

where $\hat{R}_n(\mathbf{x}, \mathcal{F})$ for a vector-valued function class $\mathcal{F}$ is defined as $\mathrm{E}[\sup_{f \in \mathcal{F}} 1/n\| \sum_{i=1}^n s_i f(x_i)\|]$ and $\sigma$ is the ReLU. As a result, we obtain

$$\hat{R}_n(\mathbf{x}, F_{1:l,\mathrm{inj}}^{\mathrm{ReLU},L}(C_1, C_2, D)) \leq 2\|W_L\|_{2,2} \hat{R}_n(\mathbf{x}, F_{1:l,\mathrm{inj}}^{\mathrm{ReLU},L-1}(C_1, C_2, D))$$
$$\leq 2^{L-l} \prod_{j=l+1}^L \|W_j\|_{2,2} \hat{R}_n(\mathbf{x}, \tilde{2}_{1:l,\mathrm{inj}}(C_2, D))$$
$$\leq 2^{L-l} \|g\|_{H_l} \bigg( \prod_{j=l+1}^L \|W_j\|_{2,2} \bigg) \bigg( \prod_{j=1}^l \frac{G_j \|K_{\sigma_j}\| \|W_j\|^{s_j}}{\det(W_j^* W_j)^{1/4}} \bigg).$$

We can also apply other peeling approaches in the same manner as the above case.

## H  EXAMPLES OF CONCRETE KOOPMAN-BASED BOUNDS

We show examples of our Koopman-based bounds.

**Example 3** *Let $g(x) = \mathrm{e}^{-c\|x\|^2}$. Let $p_j(\omega) = 1/(1 + \|\omega\|^2)^{s_j}$ for $s_j > d_j/2$. We consider a shallow network $f(x) = g(Wx + b)$. Assume $d_1 \geq d_0$ and $W$ is full-rank. The final nonlinear transformation $g$ in $f$ maps the high dimensional vector on $\mathbb{R}^{d_1}$ to a scalar value. In this case, $f_1(x) = g(x + b)$ is also the Gaussian. We have*

$$\hat{R}_n(\mathbf{x}, F_{\mathrm{inj}}(C, D)) \leq \frac{B}{\sqrt{n}} G_1 \|g\|_{H_1} \frac{\max\{1, \|W\|\}^{s_0}}{\det(W^* W)^{1/4}}.$$

*Since $\|g\|_{H_1} = \|f_1\|_{H_1}$, by Eq. (15), we have*

$$G_1^2 \|g\|_{H_1}^2 = \frac{\|f_1|_{\mathcal{R}(W)}\|_{H_{p_0(\mathcal{R}(W))}} \|g\|_{H_1}}{\|f_1\|_{H_1}} \sim c^{s_0 - d_0/2} \pi^{-2s_0 + 1} 2^{-s_0 - d_0/2} \Gamma(s_0 + d_0/2) \prod_{i=1}^{d_0-2} \tilde{c}_i.$$

*As a result, we have*

$$\hat{R}_n(\mathbf{x}, F_{\mathrm{inj}}(C, D))$$
$$\lesssim \frac{B}{\sqrt{n}} c^{s_0/2 - d_0/4} \pi^{-s_0 + 1/2} 2^{-s_0/2 - d_0/4} \bigg( \prod_{i=1}^{d_0-2} \tilde{c}_i \bigg)^{1/2} \Gamma(s_0 + d_0/2)^{1/2} \frac{\max\{1, \|W\|\}^{s_0}}{\det(W^* W)^{1/4}}.$$

*Note that $\dim(\mathcal{R}(W)) = d_0$ is the dimension of the input and $s_0$ is chosen as $s_0 > d_0/2$. They are independent of the structure of the network.*

**Example 4** *Let $\sigma(x) = (\mathrm{e}^{-c_1\|x\|^2}, \ldots, \mathrm{e}^{-c_{d_1}\|x\|^2})$. Let $p_j(\omega) = 1/(1+\|\omega\|^2)^{s_j}$ for $s_j > d_j/2$. We consider a shallow network $f(x) = W_2\sigma(W_1 x + b)$. Assume $d_1 \geq d_0$, $d_2 = 1$ and $W$ is full-rank. Using the "peeling" approach and Example 3, we obtain*

$$\hat{R}_n(\mathbf{x}, F^2_{1:1,\mathrm{inj}}(C_1, C_2, D)) \leq \|W_2\|_{2,2}\hat{R}_n(\mathbf{x}, F^1_{\mathrm{inj}}(C_2, D))$$

$$\lesssim \|W_2\|_{2,2}\sum_{i=1}^{d_1}\frac{B}{\sqrt{n}}c_i^{s_0/2-d_0/4}\pi^{-s_0+1/2}2^{-s_0/2-d_0/4}\left(\prod_{i=1}^{d_0-2}\tilde{c}_i\right)^{1/2}\Gamma(s_0+d_0/2)^{1/2}\frac{\max\{1,\|W_1\|\}^{s_0}}{\det(W_1^*W_1)^{1/4}}.$$

*Here, we used the inequality*

$$\hat{R}_n(\mathbf{x}, \mathcal{F}) = \frac{1}{n}\mathrm{E}\left[\sup_{f\in\mathcal{F}}\left\|\sum_{i=1}^n s_i f(x_i)\right\|\right] = \frac{1}{n}\mathrm{E}\left[\sup_{f\in\mathcal{F}}\sqrt{\sum_{j=1}^d\left(\sum_{i=1}^n s_i(f(x_i))_j\right)^2}\right]$$

$$\leq \frac{1}{n}\mathrm{E}\left[\sup_{f\in\mathcal{F}}\sum_{j=1}^d\sqrt{\left(\sum_{i=1}^n s_i(f(x_i))_j\right)^2}\right] = \frac{1}{n}\mathrm{E}\left[\sup_{f\in\mathcal{F}}\sum_{j=1}^d\left|\sum_{i=1}^n s_i(f(x_i))_j\right|\right]$$

$$\leq \frac{1}{n}\sum_{j=1}^d\mathrm{E}\left[\sup_{f\in\mathcal{F}}\left|\sum_{i=1}^n s_i(f(x_i))_j\right|\right],$$

*where $(f(x_i))_j$ is the jth element in the vector $f(x_i)$. In this case, the bound depends on $d_1$ linearly.*

# I  INJECTIVITY OF $\tilde{W}_j$

The operator $\tilde{W}_j$ defined in Subsection 4.3.1 is injective. Indeed, assume $(W_1 x, P_1 x) = (W_1 y, P_1 y)$ for $x, y \in \mathbb{R}^{d_0}$. Then, we have $x - y \in ker(W_1)$. On the other hand, we have $P_1(x-y) = 0$. Since $x - y \in ker(W_1)$, we have $x - y = P_1(x-y) = 0$. The case of $j \geq 2$ is the same.

# J  EXPERIMENTAL DETAILS AND ADDITIONAL EXPERIMENTAL RESULTS

We explain details of experiments in Section 5 and show additional experimental results. All the experiments were executed with Python 3.9 and TensorFlow 2.6.

## J.1  VALIDITY OF THE BOUND (SYNTHETIC DATA)

We constructed a network $f_\theta(x) = g(W_2\sigma(W_1 x + b_1) + b_2)$, where $W_1 \in \mathbb{R}^{3\times3}$, $W_2 \in \mathbb{R}^{6\times3}$, $b_1 \in \mathbb{R}^3$, $b_2 \in \mathbb{R}^6$, $\theta = (W_1, b_1, W_2, b_2)$, $\sigma(x) = ((1+\alpha)x + (1-\alpha)x\,\mathrm{erf}(\mu(1-\alpha)x))/2$, and $g(x) = \mathrm{e}^{-\|x\|^2}$. Here, erf is the Gaussian error function, and $\sigma$ is a smooth version of Leaky ReLU proposed by Biswas et al. (2022). We set $\alpha = \mu = 0.5$. For training the network, we used $n = 1000$ samples $x_i$ ($i = 1, \ldots, 1000$) drawn independently from the normal distribution with mean 0 and standard deviation 1. The weight matrices are initialized by Kaiming Initialization (He et al., 2015), and we used the SGD for the optimizer. In addition, we set the error function as $l_\theta(x, y) = |f_\theta(x) - y|^2$, and added the regularization term $0.01(\prod_{j=1}^2 \det(W_j^*W_j)^{-1/2} + 10\prod_{j=1}^2\|W_j\|)$. We added this regularization term since, according to our bound, both the determinant and the operator norm of $W_j$ should be small for achieving a small generalization error. The generalization error here means $|\mathrm{E}[l_\theta(x, t(x))] - 1/n\sum_{i=1}^n l_\theta(x_i, t(x_i))|$, which is compared to our bound $O(\prod_{j=1}^L\|W_j\|^{s_j}/(\det(W_j^*W_j)^{1/4}))$ in Figure 1 (a). Here, we set $s_j = (d_j + 0.1)/2$.

## J.2  VALIDITY OF THE BOUND (MNIST)

We constructed a network $f_\theta(x) = g(W_4\sigma(W_3\sigma(W_2\sigma(W_1 x + b_1) + b_2) + b_3) + b_4)$ with dense layers, where $W_1 \in \mathbb{R}^{1024\times784}$, $W_2 \in \mathbb{R}^{2048\times1024}$, $W_3 \in \mathbb{R}^{2048\times2048}$, $W_4 \in \mathbb{R}^{10\times2048}$, $b_1 \in \mathbb{R}^{1024}$, $b_2 \in \mathbb{R}^{2048}$, $b_3 \in \mathbb{R}^{2048}$, $b_4 \in \mathbb{R}^{10}$, $\theta = (W_1, b_1, W_2, b_2, W_3, b_3, W_4, b_4)$, $\sigma$ is the same function as Section J.1, and $g$ is the softmax. See Remark 1 for the validity of our bound for the case where $g$ is the softmax. For training the network, we used only $n = 1000$ samples to create a

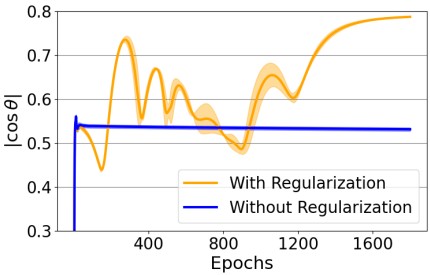

Figure 2: Behavior of the value $|\cos(\theta)|$. Here, $\theta$ is the maximum value of the angles between the output of the second layer and the directions of singular vectors of $W_3$ associated with the singular values that are larger than $0.1$.

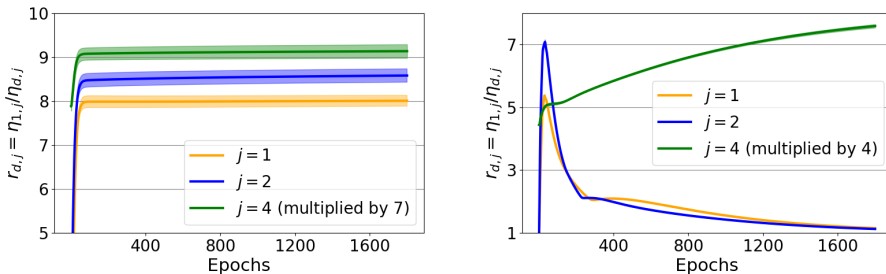

Figure 3: The ratio $r_{d,j} = \eta_{1,j}/\eta_{d,j}$ of singular values (condition number) of weight matrices for layers $j = 1, 2, 4$. (Right) Without regularization (Left) With the regularization based on our bound.

situation where the model is hard to generalize. We consider the regularization term $\lambda_1 \|W_j\| + \lambda_2/\det(I + W_j^* W_j)$, where $\lambda_1 = \lambda_2 = 0.01$ to make both the norm and the determinant of $W_j$ small (thus, makes our bound small). Based on the observation in the last part of Subsection 4.4, we set the regularization term for only $j = 1, 2$. The weight matrices are initialized by the orthogonal initialization for $j = 1, 2$ and by the samples from a truncated normal distribution for $j = 3, 4$, and we used Adam (Kingma & Ba, 2015) for the optimizer. In addition, we set the error function as the categorical cross-entropy loss.

### J.2.1 TRANSFORMATION OF SIGNALS BY LOWER LAYERS

We also investigated the transformation by lower layers, as we stated in the last part of Subsection 4.4. We computed $|\cos(\theta)|$, where $\theta$ is the maximum value of the angles between the output of the second layer and the directions of singular vectors of $W_3$ associated with the singular values that are larger than $0.1$. The results are illustrated in Figure 2. We can see that with the regularization based on our bound, as the test accuracy grows, the value $|\cos(\theta)|$ also grows. This result means that the signals turn to the directions of the singular vectors of the subsequent weight matrix associated with large singular values. That makes the extraction of information from the signals in higher layers easier. On the other hand, without the regularization, neither the test accuracy nor the value $|\cos(\theta)|$ do not become large after a sufficiently long learning process (see also Figure 1 (b)). The results in Figures 1 (b) and 2 are obtained by three independent runs.

### J.3 SINGULAR VALUES OF THE WEIGHT MATRICES

We constructed an AlexNet Krizhevsky et al. (2012) where the ReLU activation function is replaced by the smooth version of Leaky ReLU ($\sigma$ in Section J.1) to meet our setting. For training the network, we used $n = 50000$ samples and used the Adam optimizer. We set the error function as the categorical cross-entropy loss. We show the test accuracy through the learning process in Figure 4. In addition to the AlexNet, we also computed the ratio $r_{d,j}$ of the largest and the smallest singular values (condition number) of the weight matrices for the network we used in Section J.2. Since the

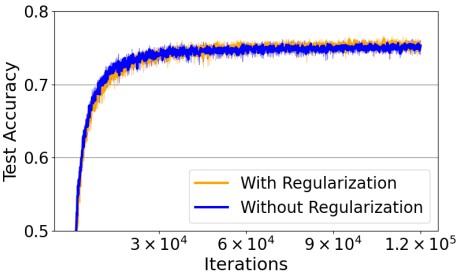

Figure 4: Test accuracy of AlexNet traind by CIFAR-10 with and without regularization.

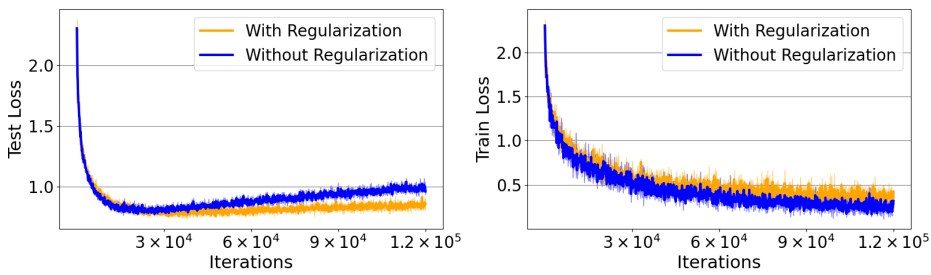

Figure 5: Test and train loss of AlexNet trained by CIFAR-10 with and without regularization. (Right) Test loss (Left) Train loss.

behavior of the singular values of $W_3$ was unstable and did not have clear patterns, we only show the result for $j = 1, 2, 4$ in Figure 3. We scaled the values for $j = 4$ for readability. In the case of the AlexNet, the behavior of singular values of each weight matrix was different depending on the layer. However, in the case of the network in Section J.2, without the regularization, the condition number $r_{d,j}$ stagnates for $j = 1, 2, 4$ through the learning process, and the test accuracy also stagnates. On the other hand, with the regularization based on our bound, $r_{d,j}$ becomes small for $j = 1, 2$ as the learning process proceeds by virtue of the regularization. We can also see that $r_{d,j}$ grows for $j = 4$ as the learning process proceeds, which makes the angle $\theta$ in Figure 2 large. As discussed in the last part of Subsection 4.4, we can conclude that the regularization transforms the signals in lower layers ($j = 1, 2$) and makes it easier for them to be extracted in higher layers ($j = 4$), and the test accuracy becomes higher than the case without the regularization. The results in Figures 1 (c), 3, and 4 are obtained by three independent runs.

### J.4 VALIDITY OF THE BOUND (CIFAR-10)

We used the same network and the same dataset as Appendix J.3 and observed the generalization property with and without a regularization based on our result. We consider the regularization term $\lambda_1 \|W_j\| + \lambda_2 / \|0.01I + W_j^* W_j\|$, where $\lambda_1 = 0.1$ and $\lambda_2 = 0.001$ to make both the largest and the smallest singular values of $W_j$ small (thus, makes our bound small). Since the AlexNet is composed of convolutional layers, we represented the convolutional layers as matrices. For the convolution $\sum_{i=1}^{n} \sum_{j=1}^{m} f_{k-i,l-j} x_{k,j}$ with a convolutional filter $F = [f_{i,j}]$, we can construct a tensor $\tilde{W}_{i,j,k,l} = f_{k-i,l-j}$. If $i$ or $j$ is out of the bound of the index of the filter, then we set $f_{i,j} = 0$. We can combine the indices $(i, j)$ and $(k, l)$ in $\tilde{W}_j$ and obtain a matrix $W_j$ that represents the convolution. Note that since the dimension of $W_j$ is large, setting a regularization term with the determinant of $W_j$ can cause numerical overflows. Thus, we set $\|0.01I + W_j^* W_j\|$ instead of its determinant in the same manner as Appendix J.2. Based on the observation in the last part of Subsection 4.4, we set the regularization term for only $j = 1, 2$. Figure 4 shows the test accuracy obtained with and without the regularization. The behavior of the accuracy obtained with and without the regularization are similar. Figure 5 shows the test and train loss. We can see that without the regularization, although the train loss becomes small, the test loss becomes large as the iteration proceeds. On the other hand,

with the regularization, the train loss becomes small, and the test loss does not become so large as the case without the regularization.

## K  NORM OF THE SOBOLEV SPACE

Let $p(\omega) = (1 + \|\omega\|^2)^s$ with $s \in \mathbb{N}$. We can represent the Sobolev norm $\|f\|_{H_p(\mathbb{R}^d)}$ using the derivatives of $f$ if $s \in \mathbb{N}$. Indeed, we have

$$
\begin{aligned}
\|f\|_{H_p(\mathbb{R}^d)}^2 &= \int_{\mathbb{R}^d} |\hat{f}(\omega)|^2 (1 + \|\omega\|^2)^s \mathrm{d}\omega = \int_{\mathbb{R}^d} |\hat{f}(\omega)|^2 \sum_{i=0}^s \binom{s}{i} \|\omega\|^{2i} \mathrm{d}\omega \\
&= \int_{\mathbb{R}^d} |\hat{f}(\omega)|^2 \sum_{i=0}^s \binom{s}{i} \left( \sum_{j=1}^d \omega_j^2 \right)^i \mathrm{d}\omega \\
&= \int_{\mathbb{R}^d} |\hat{f}(\omega)|^2 \sum_{i=0}^s \binom{s}{i} \sum_{|\alpha|=i} \binom{i}{\alpha} (\omega^\alpha)^2 \mathrm{d}\omega = \sum_{|\alpha| \le s} \frac{s!}{(s-|\alpha|)! \alpha!} \int_{\mathbb{R}^d} |\hat{f}(\omega) \omega^\alpha|^2 \mathrm{d}\omega \\
&= \sum_{|\alpha| \le s} \frac{s!}{(s-|\alpha|)! \alpha!} (2\pi)^d \|\partial^\alpha f\|_{L^2(\mathbb{R}^d)}^2 .
\end{aligned}
$$

