# OpenReview forum: "Koopman-based generalization bound: New aspect for full-rank weights"
_ICLR.cc/2024/Conference — ICLR 2024 poster_

### Official Review · Reviewer_ebFP · 2023-10-29

**Soundness:** 2 fair
**Presentation:** 2 fair
**Contribution:** 3 good
**Rating:** 6
**Confidence:** 3

**Summary:**

This paper proves generalization bounds for deep neural networks by establishing that they belong to the RHKS of sobolev spaces of a given order: in the most basic result (theorem 1 and proposition 5), a very concrete bound is shown which under the assumption that the transformations associated to the weights in each layer are invertible. The bound scales as the product of the $s$th power of the operator norms of the weights divided by the square root of their determinant. There is also a factor  $\|K_{\sigma}\|_H$ which depends on the activation function. Furthermore, follow-up theorems extend the results to the situation where the weight matrices are not invertible but only injective: in this case, a similar bound holds with the determinant of the weight replaced by the square root of the determinant of $W^{adj} W$, but at the cost of a product of factors $G_j$, which depend on the isotropy of the networks' function. Finally, further generalizations are provided which completely circumvent the need for injectivity by considering an augmented version of the network where each layer outputs a copy of its input (in addition to outputting the usual output). It is also shown that the bounds can be combined with existing bounds by employing one method for some of the layers and another for the remaining layers. Experiments demonstrate that imposing regularization inspired from the bounds provided improves performance.

**Strengths:**

This is an **extremely interesting direction** which, to the best of my knowledge, is underexplored. The results presented in this paper have the potential to be of great interest to the community for further research, since it approaches the problem from an entirely different perspective: instead of controlling the function class capacity through norms of the weights or number of parameters, properties of the learned functions in terms of their smoothness over the inputs are used instead. This may really be a key to more a more satisfying approach to generalization bounds in the overparametrized setting.


Given the great potential this work has, I am quite disappointed that the treatment offered doesn't more thoroughly study in a reader-friendly way the concrete implications in terms of lack of architectural dependencies for concrete networks. Fortunately, ICLR allows authors to upload a new pdf with very substantial revisions, so I am looking forward to that and may increase my score to 8 if my doubts are all thoroughly resolved.

**Weaknesses:**

The writing is quite crisp and abstract, sometimes to the detriment of precision. I think the results make sense in that bounds for the norms of the neural network are indeed given, but the interpretation in terms of asymptotics and the lack of dependence on the number of parameters don't fully make sense due to the presence of obscure quantities with unclear asymptotic behavior.

 To be honest, I am **not completely convinced of the correctness** of the final conclusions from a mathematical standpoint. In particular, the notations used by the authors rely a lot on Sobolev norms which are then absorbed into the O notations as if they were constants, and this process is not performed nearly carefully enough to ensure that no additional dependency on the number of parameters appears.  Here are a few examples where there are imprecisions which must be very thoroughly cleared up during the rebuttal for me to keep my score or increase it:

1 (may be fixable): Proposition 5 includes factors of $\|K_{\sigma}\|$. According to the authors, such factors can be controlled by Proposition 1, which seems to be the justification for absorbing those factors into the $O$ notation in equation (1). I can sort of believe that the final conclusion, but note that the proof given is at best sloppy: each term in the sum over multi-indices with components living in a set of cardinality equal to the layer width is bounded individually by a term $C_{\beta,\gamma,\delta}$, thus, this analysis cannot be used directly to obtain a result truly of the order claimed in equation (1), since the use of this proposition would introduce **an additional dependency on the width of the network**. Now, I agree that this is probably avoidable by using a **component-wise** activation function, which would make most of the terms cancel in the sum,  but this assumption is not even clearly stated, nor is it used anywhere. There are many missing details for the results to truly qualify as applicable to concrete neural networks.


2. (more serious) I am not convinced by the applicability of the result in the case of injective but non bijective maps: the bound contains the factors $G_j$, which depend on the "the isotropy of $f_j$", with $f_j$ only definable via the composition of the network's functions. It is absolutely unclear to what extent these quantities can be considered as constants. Even in the extremely hypothetical case where $G_j\leq G$ for some absolute constant $G$, there is still an exponential dependence in the depth of the network which is not explicitly written in the $O$ notation. In addition, it doesn't seem to be the case that the $G_j$ can be controlled properly either. In page 6, below lemma 8, the authors attempt to reassure the readers by giving an example where $G_j$ can be bounded by $(4/\pi)^{dim(R(W_j)^\top)/4}$. Note that even in that case, **this term introduces intractable, exponential dependence** in the architectural parameters $L$ (the depth) and $dim(R(W_j)^\top)$ (which is closely related to the width).


3. Similarly, the argument in appendix B is a bit vague and certainly appears to introduce dimensional dependence.







==============Minor comments (maths)=======


In general, it would be nice to make the paper more reader-friendly by adding more lines in the calculations for readers not familiar with the techniques used. For instance, it would be nice to remind the readers in a separate theorem of the Faa di Bruno formula mentioned on page 13 at the beginning of the proof of Proposition 1.


For instance, in the proof of Theorem 2 on page 14, the fact that the $s_i$s are Rademacher variables is not even explicitly stated.

The implication of Assumption 2 should be explained in terms of concrete assumptions about the network and the inputs to the network. I know that later theorems rely on a concrete formula for $p(\omega)$ which appears to make this assumption hold trivially, but this absolutely should be explained explicitly. It also seems like not choosing the $p(\cdot)$ earlier on

The spaces $R(W_j)$, which refer to the column spaces of $W_j$, should also be defined somewhere. The same applies to the non-standard notation $W^{-\star}$, which refers to the inverse of the adjoint of $W^{*}$. It also seems a little strange to use conjugate transpose notation (without introduction or justification) when all the weights are presumably real).

In addition, some of the basic notation for Fourier analysis and the relevant inner products absolutely should be included. Note that various authors use different constants in the definition of the Fourier transform, and it is not clearly stated in this paper which convention is used. I deduced from the first line of the proof of Lemma 3 on page 13 the convention used is the one where there is no constant factor in the definition of the Fourier transform. Similarly, the second equality on the same line takes some time to digest without additional details.

Similarly, the fact that the Sobolev norms are both equal to sums over all multi-indices of the relevant derivatives, and to the RKHS norm defined at the bottom of page 3 should be explained in much more detail.

Also, the main paper and the appendix are too heavily reliant on each other, it is better practice to make the appendix fully mathematically self-contained, which would imply at a minimum the following reminders to the readers: the definition of $J\sigma^{-1}$ on page 13, a reminder of the definitions of $F_{inj}$ on page 14 in the proof of Theorem 6, a better separation go Proposition 11 from proposition 10 on page 16 (since it actually refers to a completely different setting), and the definition of $G_j$ at the beginning of page 15.





============minor typos/ language=======


Fourth line of the introduction: "a large number of parameters make the complexity" should be "a large number of parameters makes the complexity"

at the beginning of remark 1, it would be better to write " Let $g$ be a smooth function which doesn't decay at infinity (e.g., a sigmoid), ..."

In the beginning of Section 4.3, "as a result $h\circ W_j$ does not contained in..." should be "as a result $h\circ W_j$ is not contained in..."

Section 4.3.2 "We only need...., not whole $W_j$" should be "We only need...., not the whole of  $W_j$"


Just before the second equation in Section 4.4 " set of all functions which has" ==> "set of all functions which have"

Page 9 "we constructed a network and learned it" ===> "trained"


Top of Page 18 in Appendix B, there shouldn't be a capital letter at  "then By"

**Questions:**

What is the norm of the bump functions $\phi_j$ assumed to satisfy in Proposition 10? Are they constant, or equal to 1? How can this be achieved without additional dependence on the ambient dimension at all? Could you provide a concrete example?

Could you address the points mentioned in the weaknesses, especially the control over the quantity $G_j$ (from lemma 8)and avoiding dimensional dependence when summing over multi-indices in Sobolev norms?

Could you write a complete and detailed version of your result for a fully concrete example where the loss function is either the cross entropy loss or the square loss and the activation is component-wise and fully explicit (e.g. the smooth version of leaky relu), without using any undefined quantity such as $G_j$, $f_j$ or even $\phi_j$ (you can choose a concrete $\phi_j$ if necessary, and show that the corresponding factors in the bound are bounded by the absolute constants)?  How can you concretely control the quantity $G_j$ in the case where we have a two-layer neural network with a very wide hidden layer?

---

> ### Author Response · Authors · 2023-11-19
>
> Thank you for your very constructive comments.
> As you pointed out, some of our results are not properly stated, or we need more clear explanations about the factors that appeared in our results.
> Analyzing them for completely general cases is challenging, but we tried to give examples and make readers easily understand as much as possible.
> Although our analysis may have room to improve, we emphasize that our main contribution is introducing an operator theoretic approach to shed light on why networks with high-rank weight matrices generalize well.
> Our results provide a new direction for analyzing generalization of networks.
> Below, we address each comment.
> We revised the paper based on your comments.
> The revised parts are colored in red.
>
> **Norm of the bump function**
>
> As you pointed out, the $O$ notation in Proposition 10 was improperly used in the original version.
> We updated Proposition 10 and its proof without using the $O$ notation.
> The norm of the bump function depends on the support of the function and parameters in the function.
> For example, we can use the bump function $\psi$ defined in Appendix D.
> Unfortunately, upper bounding the norm of this bump function analytically is challenging.
> However, we have the following observation.
> If the parameters $a$ and $b$ are large, then the support of $\psi$ becomes large, which makes the $L^2$-norm of $\psi$ large, and the Sobolev norm of $\psi$ also becomes large.
> If $a-b$ is small, then $\vert \psi(x)-\psi(y)\vert/\Vert x-y\Vert$ for $\Vert x\Vert^2=a$ and $y=(b/a)x$ becomes large.
> Thus, the $L^2$-norms of the derivatives of $\psi$ are expected to be large, and the Sobolev norm of $\psi$ also becomes large if $a-b$ is small.
> Please note that in Proposition 10, we only need the norm of $\psi_j$ on $ker(W_j)$.
> Thus, it depends only on $r_j=dim(ker(W_j))$, but $r_j$ is not directly related to $d_j$ and if $W_j$ is high-rank, $r_j$ is small.
> Moreover, combining with the factor $G_j$, the norm of $\psi_j$ can be canceled.
> Please see Remark 9 and Appendix F for details.
>
> We can also apply the above bump function to Proposition 9, which involves $\Vert \tilde{g}\Vert=\Vert \psi\Vert\,\Vert g\Vert$.
> In this case, according to the original version of Subsection 4.3.1, the above $d$ should be $d=\sum_{j=0}^{L-1}d_j$, and the norm of $\psi$ depends on $d$.
> However, we can alleviate the dependency on the width $d_j$ by modifying the definition of $\tilde{W}\_j$ using the projection $P_j$ onto $ker(W_j)$ .
> Please see the modified version of Subsection 4.3.1 and Appendix I.
> We define $\tilde{g}$ in the same manner as that in Subsection 4.3.1.
> The norm of $\psi$ depends on $\sum_{k=0}^{L-1}r_k$, but $r_j=\mathrm{dim}(ker(W_j))$ is not directly related to $d_j$.
> If $r_j$ is small, then $\sum_{k=0}^{L-1}r_k$ is also small.
>
> We emphasize that our bound focuses on the case where the weight matrices are high-rank.
> Our bound is effective in the case where $dim(ker(W_j))$ is small.
> Our fundamental result is Theorem 2 for invertible weight matrices.
> The norm of the bump function and the factor $G_j$ measure how much the situation is different from the case of invertible weight matrices.
> We also emphasize that, as we stated in Subsection 4.4, we can combine our bound with existing bounds.
> We can use existing bounds for low-rank weight matrices to obtain a tighter bound.
>
> **Factor $G_j$**
>
> We found that we can replace $G_j$ in Theorem 6 by a smaller value by considering $H_{p_{j-1}}(\mathcal{R}(W_j))$ instead of $H_{p_{j}}(\mathcal{R}(W_j))$.
> We remark that since we assume $W_j$ is injective, $dim(\mathcal{R}(W_j))=d_{j-1}$ and $dim(\mathcal{R}(W_j)^{\perp})=d_j-d_{j-1}$.
> Please see the revised version of our paper for the details of the modification of the proof.
>
> The modified version of $G_j$ does not seriously affect the bound.
> Indeed, in the case of $f_j(x)=\mathrm{e}^{-c\Vert x\Vert^2}$, we can evaluate $G_j$.
> $G_j$ becomes small as $c$ becomes large and $s_j$ and $d_j$ becomes large.
> Please see Remark 5 and Appendix C for details.

---

> > ### Author Response · Authors · 2023-11-19
> >
> > **Factor $\Vert K_{\sigma_j}\Vert$**
> >
> > To derive a bound of $\Vert K_{\sigma_j}\Vert$,
> > we bound $\sum_{\vert \alpha\vert\le s}c_{\alpha,s,d}\Vert \partial^{\alpha}(h\circ\sigma)\Vert^2$ by $\sum_{\vert \alpha\vert\le s}c_{\alpha,s,d}\Vert\partial^{\alpha}h\Vert^2$.
> > As the proof of the boundedness of $\Vert K_{\sigma}\Vert$, one strategy is using the Faa di Bruno formula.
> > However, as we added explanations in Appendix B, this strategy does not give us a tight bound since it involves a summation of derivatives, and the bound depends on the number of terms in the sum.
> > Moreover, estimating the number of terms in the sum is difficult.
> >
> > Our main goal of this paper is to investigate how the property of the weight matrices affects the generalization property.
> > Since the norm of $K_{\sigma}$ does not depend on the weight matrices, if we assume the structure of the network is given, the property of the weight matrices does not affect the norm of $K_{\sigma}$.
> > As we stated in Conclusion, investigating $K_{\sigma}$ and deriving a tighter bound is future work.
> >
> > In addition, the dependency of $\Vert K_{\sigma}\Vert$ on $d_j$ and $s_j$ can be alleviated to $d_{j-1}$ and $s_{j-1}$ by the factor $G_j$.
> > Please see Remark 5 and Appendix C for more details.
> > We admit that if the network is deep and if the width is sequentially increased, then the factor $G_j$ does not remove the dependency on the width of the intermediate layers.
> > In this work, we considered the Koopman operator for each transformation.
> > However, combining several layers that have similar roles together and considering the Koopman operator for the combined transformation may alleviate the dependency on the width of the intermediate layers.
> > In our future work, we will try to derive a tight bound by combining several layers together.
> >
> > **Clarification of Appendix B (Appendix E in the current version)**
> >
> > We updated Appendix E and added the explanation of the derivative of $\psi$.
> > Assume $\delta_L=1$, $\psi(x)=1$ for $x\in\Omega$, and $\Omega$ is an interval (e.g., $\psi$ is the bump function defined in Appendix D).
> > Then we can create a new function $\tilde{\psi}$ that satisfies $\Vert \tilde{\psi}^{(i)}\Vert_{L^2(\mathbb{R})}=\Vert \psi^{(i)}\Vert_{L^2(\mathbb{R})}$ for any $i=1,2,\ldots$, from $\psi$.
> > Here, $\psi^{(i)}$ is the $i$th derivative of $\psi$.
> >
> > **Concrete bound**
> >
> > We added a section for showing concrete examples of our bounds in Appendix (Appendix H).
> > In Example 3, we consider a shallow network.
> > In this case, the bound depends only on the input dimension $d_0$.
> > In Example 4, we combine our bound with an existing bound and obtain a bound of the network with an additional weight matrix to the one considered in Example 3.
> > In this case, the bound depends on $d_1$ linearly.
> > Please note that our bound does not depend on the loss function.
> >
> > **Minor comments**
> >
> > Thank you for your comments.
> > We revised the paper based on your comments.

---

> > ### Comment · Reviewer_ebFP · 2023-11-21
> > **Rotational symmetry**
> >
> > Dear Authors,
> >
> >
> > First of all, thank you so much for the detailed revision. I am working my way through the details as best as possible.  I especially appreciate the fact that you have added additional details in some of the mathematical derivations as I have requested.
> >
> > I do have a couple more questions, the first one of which concerns the $G_j$ and the additional section Appendix H with the two layer example.  Apologies if I misunderstood something, please kindly provide more explanations.
> >
> > I appreciate the discussion of the calculation of $G_j$ and the admission that "your, the main goal of this paper is to investigate how the property of the weight matrices affects generalization" (without regards to existing dependence on architectural parameters). This is an **enormous restriction** that severely restricts the applicability of your results, but if you had been able to completely circumvent this difficulty, the work would have been nothing short of exceptional.
> >
> >
> >
> >
> > So my **first (most important) question is**: in remark 5 (appendix C), at the second equality in the calculation of $G_j$, you are using the rotational symmetry of the Gaussian function $f_j$. **But I don't see how this could possibly hold when there are several layers.** Could you explain a bit more whether this step holds in general without having to assume something about $f_j$ (which is a little bit like cheating) and instead relying only on a definition of $g$?
> >
> >
> >
> > **Section question**: In Section H where you consider the case of a two layer neural network (example 3), the same thing bothers me a bit: is your network 2-layer or 1-layer (assuming we consider $g$ as the loss function)? There seems to be some confusion as to the role of $\tilde{f}$ versus $g$.
> >
> > My final question in this post relates to example 4: I am very happy that your conclusion is that there is dependence on the input dimension (but not on the hidden dimension). **This is exactly what I would expect at an intuitive level and what makes your approach unique! **
> >
> > Could you write down the detailed proof using the "peeling technique"? The current version still looks more like a sketch. Also, what is $g$ in this case? Will the result hold with the square loss? How about a classification context with the cross entropy or a margin based loss?

---

> ### Comment · Reviewer_ebFP · 2023-11-21
> **Sobolev norms**
>
> Thanks for adding the "note" under example 1. It really will help improve the readability of the paper. In addition, could you also provide a publicly available reference with a specific page and theorem number?

---

> ### Comment · Reviewer_ebFP · 2023-11-21
> **RELU**
>
> Dear Authors,
>
> I have one more open ended question.
>
> It seems that a lot of the issues that make the results hard to improve in terms of dimensional dependence comes from the application of the Faa di Bruno formula for the higher derivatives. Thanks for your detailed explanation! It took me a while but thanks to your help I finally understand that even if the activation function is elementwise, we still have  a problem with dimensional dependence. However, I am wondering if this can be fixed by using RELU: in know that we can no longer apply the Faa di Bruno formula at fact value, but it seems like smoothness at downstream layers will counteract the lack of smoothness of RELU near zero: for instance, if f is sufficiently smooth, f(RELU (x)) should become smooth as well. In addition, this would make the cross terms cancel. Can you try this out or explain why it wouldn't work?

---

> > ### Author Response · Authors · 2023-11-22
> >
> > Dear Reviewer ebFP,
> >
> > Thank you very much for checking the rebuttal and the modified version of the paper.
> > Below, we provide additional comments.
> > We also revised the paper.
> > The revised parts are colored in blue.
> > We hope these comments answer your questions and that the revision gives readers more information.
> >
> > **Factor $G_j$**
> >
> > We investigated $G_j$ in more detail.
> > We first obtain a result that shows the average of $G_j$ is bounded by a constant that is independent of $f_j$.
> > Please see Appendix C for more details.
> > In this result, the upper bound depends seriously on $s_j$ and $d_j$.
> > Also, we need an assumption that $f_j$ does not have its support near $0$.
> > We admit that this bound is not tight, and we need an assumption regarding the support of $f_j$.
> > However, the fact that $G_j$ is bounded by a constant that is independent of $f_j$ is not trivial.
> >
> > In addition, we can generalize the result about the upper bound of $G_j$ in the previous version.
> > The previous result was for the case where $f_j$ is the Gaussian.
> > The new result is valid for $\hat{f}_j$ that decays slower or equal to the speed of the Gaussian in the direction of $\mathcal{R}(W_j)^{\perp}$.
> >
> > For now, we need some assumptions regarding $f_j$ to derive an upper bound of $G_j$.
> > Removing these assumptions and dealing with more general cases are future work.
> > However, we believe that the above additional results help readers understand the property of $G_j$.
> >
> > There were small errors in Appendices C and H.
> > We corrected them.
> > They do not change the dependence of the final results on $s_j$ and $d_j$.
> > Since the factor is complicated, for simplicity, we left only the asymptotically essential terms in the final result.
> >
> > **About Example 3**
> >
> > We basically consider $g$ as the nonlinear transformation in the last layer.
> > In this case, the network $f$ in Example 3 is a 2-layer network.
> > Our result is just about the Rademacher complexity of the function class of neural networks and is independent of the loss function.
> > For connecting our results to generalization error, if the loss function is Lipschitz continuous, then we can use, e.g., Corollary 1 by Maurer, ALT 2016.
> > For example, the squared error $L_i(x)=\vert x-y_i\vert^2$ is not Lipschitz continuous on $\mathbb{R}$.
> > However, in practical situations, we do not need to consider the whole space for the loss function.
> > For example, if we consider the classification task, the output of the network is always on $[0,1]$, and the label $y_i$ are also in $[0,1]$, then $L_i$ is Lipschitz continuous on $[0,1]$.
> >
> > On the other hand, we can also consider $g$ as a loss function, as you pointed out.
> > In this case, the network $f$ in Example 3 is a 1-layer network.
> >
> > As for the relationship with $\tilde{f}$, we are sorry that the explanation was misleading.
> > We just wanted to explain that the network $f=g(Wx+b)$ that we are considering is similar to the standard shallow network $\tilde{f}(x)=W_2\sigma(W_1x+b)$.
> > The final transformation $g$ has a similar role to that of $W_2\sigma(\cdot)$ in $\tilde{f}$.
> > In fact, we do not need $\tilde{f}$ for obtaining the result in Example 3.
> > To avoid confusion, we removed the explanation of $\tilde{f}$ in Example 3 in the paper.
> >
> > **About Example 4**
> >
> > We added detailed explanations in Appendix G and Example 4 about the combination with the "peeling" approaches.
> > Also, the Frobenius norm was denoted as  $\Vert\cdot\Vert_{2,2}$ in the main text, but as $\Vert\cdot\Vert_{\mathrm{F}}$ in the appendix.
> > We corrected that to make the notation consistent.
> > For Example 4, $g$ is the identity function.
> > Since we used the existing result for the last layer, $g$ does not have to be in the Sobolev space.
> > Please note that as we stated above, we can use Corollary 1 by Maurer, ALT 2016, to get the result for the Rademacher complexity for Lipschitz continuous loss functions.
> >
> > **About Example 1**
> >
> > We can show the representation of the Sobolev norm with the derivatives by using the properties of the Fourier transform.
> > We added the explanation in Appendix K.
> > (To keep the section number consistent with the response to other reviewers, we put the explanation in the last part of the appendix.
> > However, we will put it in an appropriate place in the camera-ready version.)

---

> > > ### Author Response · Authors · 2023-11-22
> > >
> > > **ReLU for the activation function**
> > >
> > > Thank you very much for the idea.
> > > Focusing on the ReLU for making the situation simple is a very interesting direction for future research.
> > > Unfortunately, we cannot apply ReLU in our current setting since it is not bijective (Please see Proposition 1).
> > > Instead, we can consider the Leaky ReLU.
> > > However, since the Leaky ReLU is not smooth, it does not match our setting, either.
> > > Indeed, even if a function $f$ is sufficiently smooth, the composition of $f$ and a non-smooth function is not smooth.
> > >
> > > However, in future work, we will try to extend our results to the spaces of compactly supported functions and non-smooth functions.
> > > If we can do that, we may apply our framework to non-injective activation functions since the difficulty for the current setting comes from the non-integrability of functions composed by the non-injective activation function.
> > > Moreover, we can apply our framework to non-smooth activation functions in that case.

---

### Official Review · Reviewer_K7E7 · 2023-10-30

**Soundness:** 3 good
**Presentation:** 4 excellent
**Contribution:** 4 excellent
**Rating:** 8
**Confidence:** 4

**Summary:**

This paper establishes a new generalization bound for neural networks based on the Koopman operator. To be specific, the authors first represent the network by the product of Koopman operators. Then, new upper bounds of Rademacher complexity are derived for invertible, injective, and non-injective weight matrices, respectively. Furthermore, the Koopman-based bound is combined with other generalization bounds such that both high and low layers can be sharply bounded. Finally, numerical results validate the effectiveness of the regularization induced by the Koopman-based bound, and the different behaviors of singular values of the weight matrix for each layer are also observed.

**Strengths:**

The proposed generalization bound is sharp and fills the theoretical gap. Specifically, benefiting from the denominator induced by the Koopman operator, the generalization bound can be sharp when the condition number of the weight matrix is small. What’s more, if the weight matrices are orthogonal, the bound reduces to 1 and is independent of the width of the network. This result explains the generalization ability of neural networks when the weight matrices are full-rank.  By contrast, existing results either depend on the $(p, q)$ matrix norm, which scales by the order of $d^{1/p}$ for a $d \times d$ matrix, or become loose when faced with high-rank weight matrices.
- The authors validate the proposed bound with the help of numerical results. On one hand, experimental results on both regression and classification validate the proposed generalization bound. On the other hand, the different behaviors of singular values of the weight matrix for each layer are also observed for AlexNet on the CIFAR dataset.
- This paper is well organized, which makes it easy to understand.

**Weaknesses:**

- The authors mainly consider the neural networks with dense layers. I wonder whether these theoretical results can generalize well to neural networks with other structures such as convolution. A simple explanation is recommended.
- The experimental results on MNIST validate the effectiveness of the induced regularization term. Can it boost model performance on datasets with larger scales such as CIFAR?
- Besides, there are some typos. For example,
	- In the introduction part, "depth of the network.Another approach" should be "depth of the network. Another approach".
	- In the introduction part, the third paragraph has an extra indent.
	- In Table 1, a larger line spacing is recommended.

**Questions:**

Please refer to Weakness

**Details Of Ethics Concerns:**

nan

---

> ### Author Response · Authors · 2023-11-19
>
> Thank you for your very constructive comments.
> We addressed your comments as follows.
> We also revised the paper based on your comments.
> The revised parts are colored in red.
>
> **Generalization to other structures**
>
> We can generalize our results to convolutional layers by regarding the convolution as the action of a matrix.
> For the convolution $\sum_{i=1}^n\sum_{j=1}^mf_{k-i,l-j}x_{i,j}$ with a convolutional filter $F=[f_{i,j}]$, we can construct a tensor $W_{i,j,k,l}=f_{k-i,l-j}$.
> If $i$ or $j$ is out of the bound of the index of the filter, then we set $f_{i,j}=0$.
> We can combine the indices $(i,j)$ and $(k,l)$ and obtain a matrix that represents the convolution.
> We added the explanation of this in Appendix J.4.
> Please see Appendix J.4 for more details.
> Generalizing our results to other structures, such as pooling layer, is future work.
>
> **Experiment with CIFAR-10**
>
> We conducted an additional experiment with AlexNet and CIFAR-10 to observe the generalization property with a regularization based on our result.
> Please see Appendix J.4 for more details.
> We observed that without the regularization, although the train loss becomes small, the test loss becomes large as the iteration proceeds.
> On the other hand, with the regularization, the train loss becomes small, and the test loss does not become so large as the case without the regularization.
>
> **Other comments**
>
> Thank you for your comments.
> We revised the paper based on your comments.

---

### Official Review · Reviewer_i9ex · 2023-11-05

**Soundness:** 3 good
**Presentation:** 3 good
**Contribution:** 2 fair
**Rating:** 5
**Confidence:** 3

**Summary:**

This paper provide an operator-theoretic approach to analyzing networks. They proposed a novel bound for generalization of neural networks using Koopman operators and Rademacher complexity, which reveals a new aspect of neural networks.

**Strengths:**

1.	This paper proposed a new complexity bound that involves both the norm and determinant of the weight matrices. This bound is particularly useful when the condition numbers of the weight matrices are small.
2.	It provides a new perspective on why networks with high-rank weights generalize well. By combining our bound with existing bounds, we can obtain a more comprehensive description of the role of each layer in the network.
3.	This paper presented an operator-theoretic approach to analyzing networks, using Koopman operators to derive the determinant term in our bound. This approach offers a new way to analyze the generalization performance of neural networks.

**Weaknesses:**

This paper gives the generalization error bound of neural networks from a novel perspective which sounds very interesting and introduces new tools to generalization analysis. But since I'm not familiar with dynamic-based Koopman operators, I have some concerns that I'd like to see answered by the author.

1. As the author said, Efficient learning algorithms have been proposed by describing the learning dynamics of the parameters of neural networks by Koopman operators. It seems that the author represents the composition structure of neural networks using Koopman operators, and then uses the complexity method to give an upper bound. My question is, dynamics sound algorithm-related, while complexity is algorithm-independent, so what's the point of using Koopman operators here.

2. Looking at the entire proof, the conclusion of this paper seems to be highly related to the hypothesis of RKHS. My intuitive feeling is that the conclusion of this paper mainly comes from the RKHS assumption, and whether the neural network is abstracted into Koopman operators has little relevance. I hope the author can explain the relationship between Koopman operators, RKHS and the final conclusion and give an idea of what kind of these techniques/assumptions play a role in the proofs.

**Questions:**

See above

---

> ### Author Response · Authors · 2023-11-19
>
> Thank you for your very constructive comments.
> We addressed your comments as follows.
>
> **Reason for applying Koopman operators to generalization bound**
>
> In this study, the Koopman operators are not really related to dynamical systems.
> We use them to describe the composition structure of neural networks rather than connecting them to dynamical systems.
> By using Koopman operators, we can represent a neural network $f$ with the product of the Koopman operators: $f=K_{W_1}K_{b_1}K_{\sigma_1}\cdots K_{W_L}K_{b_L}g$, where $W_j$, $b_j$, and $\sigma_j$ are the action of the weight matrix, bias, and activation function at the $j$th layer.
> In addition, $g$ is the final nonlinear transformation.
> We can bound $\Vert f\Vert$ by the product $\Vert K_{W_1}\Vert \Vert K_{b_1}\Vert \Vert K_{\sigma_1}\Vert \cdots \Vert K_{W_L}\Vert \Vert K_{b_L}\Vert\Vert g\Vert$.
> By virtue of the representation with Koopman operators, instead of analyzing the composition structure of the neural network directly, we only need to evaluate the norm of each Koopman operator.
>
> As for the connection with algorithms, our bound provides a way of regularization (Please see Figure 1 (b) in Section 5).
> However, as of now, our analysis is independent of the existing analysis of learning dynamics with Koopman operators.
> Combining our results with the analysis of learning dynamics is an interesting direction for future work.
>
> **Relationship among Koopman operators, RKHS, and final conclusion**
>
> The derivation of our bound depends on the assumption of RKHS, but it also strongly depends on the application of the Koopman operators.
> By the assumption of RKHS, evaluating the Rademacher complexity is reduced to evaluating $\Vert f\Vert$.
> As we also stated above, by virtue of the Koopman operators, we can bound $\Vert f\Vert$ using the norm of Koopman operators.
> An important feature of our bound is that it involves the determinant of the weight matrix in the denominator.
> This determinant factor appears by bounding the norm of the Koopman operator.
> We didn't have this type of factor in existing results, which means the norm of the Koopman operator gives us a new perspective on the generalization of neural networks.
> By the effect of the determinant factor, our bound becomes small if the smallest singular value of the weight matrix is large.
> The direction of this result is completely different from that of existing results that focus on low-rank weight matrices or weight matrices with small singular values.
> Our result reveals why networks with full or high-rank weight matrices generalize well.

---

### Meta-Review · Area_Chair_pBNB · 2023-12-06

**Metareview:**

The paper provides a new generalization bound for neural networks. The approach used to produce this bound appears different than the approaches that were used to generate generalization bounds in the past. The result is interesting, and the majority of the reviewers believe the paper should be accepted.

**Justification For Why Not Higher Score:**

The derivation of a new generalization bound based on a new ingredient is most appropriate for an accept (poster) decision.

**Justification For Why Not Lower Score:**

The paper makes an interesting contribution to the ML literature.

---

### Decision · Program_Chairs · 2024-01-16

Accept (poster)